# Misfolded proteins bind and activate death receptor 5 to trigger apoptosis during unresolved endoplasmic reticulum stress

Mable Lam[1,2], Scot A Marsters[3], Avi Ashkenazi[3]*, Peter Walter[1,2]*

[1]Department of Biochemistry and Biophysics, Howard Hughes Medical Institute, University of California, San Francisco, San Francisco, United States; [2]Department of Biochemistry and Biophysics, University of California, San Francisco, San Francisco, United States; [3]Cancer Immunology, Genentech, Inc, San Francisco, United States

**Abstract** Disruption of protein folding in the endoplasmic reticulum (ER) activates the unfolded protein response (UPR)—a signaling network that ultimately determines cell fate. Initially, UPR signaling aims at cytoprotection and restoration of ER homeostasis; that failing, it drives apoptotic cell death. ER stress initiates apoptosis through intracellular activation of death receptor 5 (DR5) independent of its canonical extracellular ligand Apo2L/TRAIL; however, the mechanism underlying DR5 activation is unknown. In cultured human cells, we find that misfolded proteins can directly engage with DR5 in the ER-Golgi intermediate compartment, where DR5 assembles pro-apoptotic caspase 8-activating complexes. Moreover, peptides used as a proxy for exposed misfolded protein chains selectively bind to the purified DR5 ectodomain and induce its oligomerization. These findings indicate that misfolded proteins can act as ligands to activate DR5 intracellularly and promote apoptosis. We propose that cells can use DR5 as a late protein-folding checkpoint before committing to a terminal apoptotic fate.

*For correspondence:
ashkenazi.avi@gene.com (AA);
peter@walterlab.ucsf.edu (PW)

## Introduction

Proper folding of transmembrane and secreted proteins is critical to cell function and intercellular communication. Quality control of protein folding begins in the endoplasmic reticulum (ER) and responds to increased protein-folding demand during physiological or pathophysiological stresses. Accumulation of unfolded or misfolded proteins in the ER, known as ER stress, activates the unfolded protein response (UPR) – a network of intracellular signaling pathways that initially mount cytoprotective response to restore ER homeostasis but can ultimately switch to a pro-apoptotic program under irresolvable stress (*Walter and Ron, 2011*; *Tabas and Ron, 2011*). Two key UPR sensors, IRE1 and PERK coordinate the decision between cell survival and death through the delayed upregulation of the apoptosis-initiating protein death receptor 5 (DR5) (*Lu et al., 2014*; *Chang et al., 2018*).

During ER stress, IRE1 and PERK oligomerize upon directly binding to misfolded proteins, leading to their activation (*Karagöz et al., 2017*; *Wang et al., 2018*). PERK activation causes the selective translation of ATF4 and CHOP, which, in addition to upregulating genes that enhance the folding capacity of the ER, promotes the transcription of pro-apoptotic DR5 (*Harding et al., 2003*; *Yamaguchi and Wang, 2004*). The pro-apoptotic signal is initially counteracted by regulated IRE1-dependent mRNA decay (RIDD) that degrades DR5 mRNA (*Lu et al., 2014*). Upon prolonged ER stress, PERK exerts negative feedback on IRE1 activity attenuating RIDD, thus de-repressing DR5 synthesis to drive cell commitment to apoptosis (*Chang et al., 2018*).

**eLife digest** Proteins are chains of building blocks called amino acids, folded into a flexible 3D shape that is critical for its biological activity. This shape depends on many factors, but one is the chemistry of the amino acids. Because the internal and external environments of cells are mostly water-filled, correctly folded proteins often display so-called hydrophilic (or 'water-loving') amino acids on their surface, while tucking hydrophobic (or 'water-hating') amino acids on the inside.

A compartment within the cell called the endoplasmic reticulum folds the proteins that are destined for the outside of the cell. It can handle a steady stream of protein chains, but a sudden increase in demand for production, or issues with the underlying machinery, can stress the endoplasmic reticulum and hinder protein folding. This is problematic because incorrectly folded proteins cannot work as they should and can be toxic to the cell that made them or even to other cells. Many cells handle this kind of stress by activating a failsafe alarm system called the unfolded protein response. It detects the presence of incorrectly shaped proteins and sends signals that try to protect the cell and restore protein folding to normal. If that fails within a certain period of time, it switches to signals that tell the cell to safely self-destruct. That switch, from protection to self-destruction, involves a protein called death receptor 5, or DR5 for short. DR5 typically triggers the cell's self-destruct program by forming molecular clusters at the cell's surface, in response to a signal it receives from the exterior. During a failed unfolded protein response, DR5 seems instead to act in response to signals from inside the cell, but it was not clear how this works.

To find out, Lam et al. stressed the endoplasmic reticulum in human cells by forcing it to fold a lot of proteins. This revealed that DR5 sticks to misfolded proteins when they leave the endoplasmic reticulum. In response, DR5 molecules form clusters that trigger the cell's self-destruct program. DR5 directly recognized hydrophobic amino acids on the misfolded protein's surface that would normally be hidden inside. When Lam et al. edited these hydrophobic regions to become hydrophilic, the DR5 molecules could no longer detect them as well. This stopped the cells from dying so easily when they were under stress. It seems that DR5 decides the fate of the cell by detecting proteins that were incorrectly folded in the endoplasmic reticulum.

Problems with protein folding occur in many human diseases, including metabolic conditions, cancer and degenerative brain disorders. Future work could reveal whether controlling the activation of DR5 could help to influence if and when cells die. The next step is to understand how DR5 interacts with incorrectly folded proteins at the atomic level. This could aid the design of drugs that specifically target such receptors.

DR5 is a pro-apoptotic member of the tumor necrosis factor receptor (TNFR) superfamily that signals from the plasma membrane into the cell in response to extracellular cues (*Sheridan et al., 1997*; *Walczak et al., 1997*; *Ashkenazi, 1998*). It is constitutively expressed in various tissue types and forms auto-inhibited dimers in its resting state, analogous to other members of the TNFR family (*Spierings et al., 2004*; *Pan et al., 2019*; *Vanamee and Faustman, 2018*). In its canonical mode of activation, binding of the homotrimeric extracellular ligand TRAIL (also known as Apo2L) (*Wiley et al., 1995*; *Pitti et al., 1996*) assembles DR5 into higher-order oligomers (*Hymowitz et al., 1999*; *Mongkolsapaya et al., 1999*; *Valley et al., 2012*). Consequently, DR5 forms intracellular scaffolds in which its cytosolic death domains recruit the adaptor protein FADD and pro-caspase 8 into the 'death-inducing signaling complex' (DISC) (*Kischkel et al., 2000*; *Sprick et al., 2000*; *Jin et al., 2009*; *Dickens et al., 2012*). Upon DISC-mediated dimerization, pro-caspase 8 molecules undergo regulated auto-proteolysis to form active initiator caspase 8 (*Muzio et al., 1998*). Activated caspase 8 frequently induces the intrinsic mitochondrial apoptotic pathway by truncating Bid, a pro-apoptotic Bcl2 protein, to cause Bax-mediated permeabilization of the mitochondrial outer membrane (*Wei et al., 2001*; *LeBlanc et al., 2002*).

While DR5 and caspase 8 are both required for apoptosis during ER stress, we (*Lu et al., 2014*; *Lam et al., 2018*), along with other independent groups (*Cazanave et al., 2011*; *Iurlaro et al., 2017*; *Dufour et al., 2017*), found unexpectedly that TRAIL is dispensable for this DR5 activation. Indeed, upon ER stress, most newly synthesized DR5 molecules never make it to the plasma membrane but remain intracellular and thus inaccessible to extracellular ligands (*Lu et al., 2014*;

*Iurlaro et al., 2017*). Given that at physiological levels DR5 is auto-inhibited until activated by a ligand, it remained a mystery how DR5 is activated in response to ER stress, prompting us to interrogate its intracellular mechanism of activation.

## Results

### Misfolded proteins induce DR5-dependent apoptosis and can assemble DR5-caspase 8 signaling complexes

To examine the mechanism of cell death driven by an unmitigated protein folding burden, we induced the exogenous expression of a GFP-tagged form of the glycoprotein myelin protein zero (MPZ) in epithelial cells (*Figure 1A*). MPZ initially folds in the ER and then travels to the plasma membrane to mediate membrane adhesion in myelin-forming Schwann cells, where it is normally expressed. Mutations of MPZ that impair folding and cause its intracellular retention activate the UPR, leading to apoptosis in a manner dependent on CHOP (*Pennuto et al., 2008*). We found that in epithelial cells, titration of even non-mutant, GFP-tagged MPZ to high expression levels resulted in its intracellular accumulation, indicating a compromised MPZ folding state (*Figure 1A*). Folding-compromised MPZ induced a dose-dependent upregulation of the UPR transcriptional target genes CHOP, BiP, and DR5 (*Figure 1—figure supplement 1A*). Upregulated DR5 was retained intracellularly (*Figure 1A*, *Figure 1—figure supplement 1B*) and occurred concomitantly with cleavage of caspase 8and its downstream target caspase 3 (*Figure 1B*). By contrast, low levels of MPZ-GFP expression that exhibited proper plasma membrane localization did not induce caspase 8 or 3 activity (*Figure 1A, B*). To determine when caspase 8 became active relative to cytoprotective UPR signaling, we assessed IRE1 activity during high MPZ-GFP expression through analysis of *XBP1* mRNA splicing. As expected, IRE1-mediated *XBP1* mRNA splicing initiated a few hours post-transfection with MPZ-GFP and later attenuated (*Figure 1—figure supplement 1C*). The upregulation of DR5, caspase activity, and PARP cleavage (another indicator of apoptotic progression) occurred after the attenuation of IRE1 activity, consistent with the hallmarks of terminal pro-apoptotic UPR signaling (*Figure 1—figure supplement 1D–1E*).

To determine if DR5 was required for apoptosis during this sustained protein misfolding stress, we acutely depleted DR5 mRNA by siRNA prior to overexpressing MPZ-GFP. Knockdown of DR5 significantly reduced PARP cleavage and annexin V staining following overexpression of MPZ-GFP (*Figure 1C, D*), which was not observed in control experiments expressing cytosolic GFP. To determine if upregulation of DR5 was sufficient to induce apoptosis, we increased DR5 levels in the absence of ER stress through ectopic expression of CHOP. Comparable levels of CHOP-induced DR5 protein in the absence of ER stress drove drastically lower levels of PARP cleavage and trypan blue staining (demarking apoptotic cells) compared to the presence of misfolded-protein stress (*Figure 1—figure supplement 2A and C–D*). These results show that DR5 activation does not occur spontaneously after its upregulation but requires additional input signals conveyed by ER stress.

To assess the molecular composition of activated DR5 assemblies formed in response to ER stress, we measured caspase 8 activity in cell extracts fractionated through size exclusion chromatography. We detected increased caspase 8 activity in high-molecular w8 (MW) fractions of cells transfected with MPZ-GFP relative to GFP (*Figure 1E*). The fractions contained DR5 complexes and co-eluted with full-length MPZ-GFP but not GFP-degradation products (*Figure 1E*, lanes 2 and 4). Pull-down of DR5 from cell lysates enriched for FADD and MPZ-GFP (*Figure 1—figure supplement 3A*), suggesting that the co-elution of DR5 and MPZ-GFP in the high MW fractions resulted from their physical association. To test if MPZ physically interacted with activated DR5 complexes, we immuno-precipitated MPZ-GFP and detected DR5, FADD, and caspase 8 (both full-length p55 and its cleaved form p43) (*Figure 1F*, *Figure 1—figure supplement 3B*). Furthermore, MPZ-GFP immunoprecipitates contained 2–3-fold more caspase 8 activity compared to empty beads (*Figure 1G*, *Figure 1—figure supplement 3C*), indicating that they contained assembled DISC in a similar degree as seen after affinity purification of TRAIL-ligated DR5 (*Hughes et al., 2013*). In contrast, pull-down of cytosolic GFP did not enrich for DR5, FADD, or caspase activity (*Figure 1F, G*), confirming the selectivity for ER-folded MPZ-GFP.

To determine if misfolded proteins generally induced caspase activity through association with DR5, we overexpressed GFP-tagged forms of two other ER-trafficked proteins, rhodopsin (RHO) and

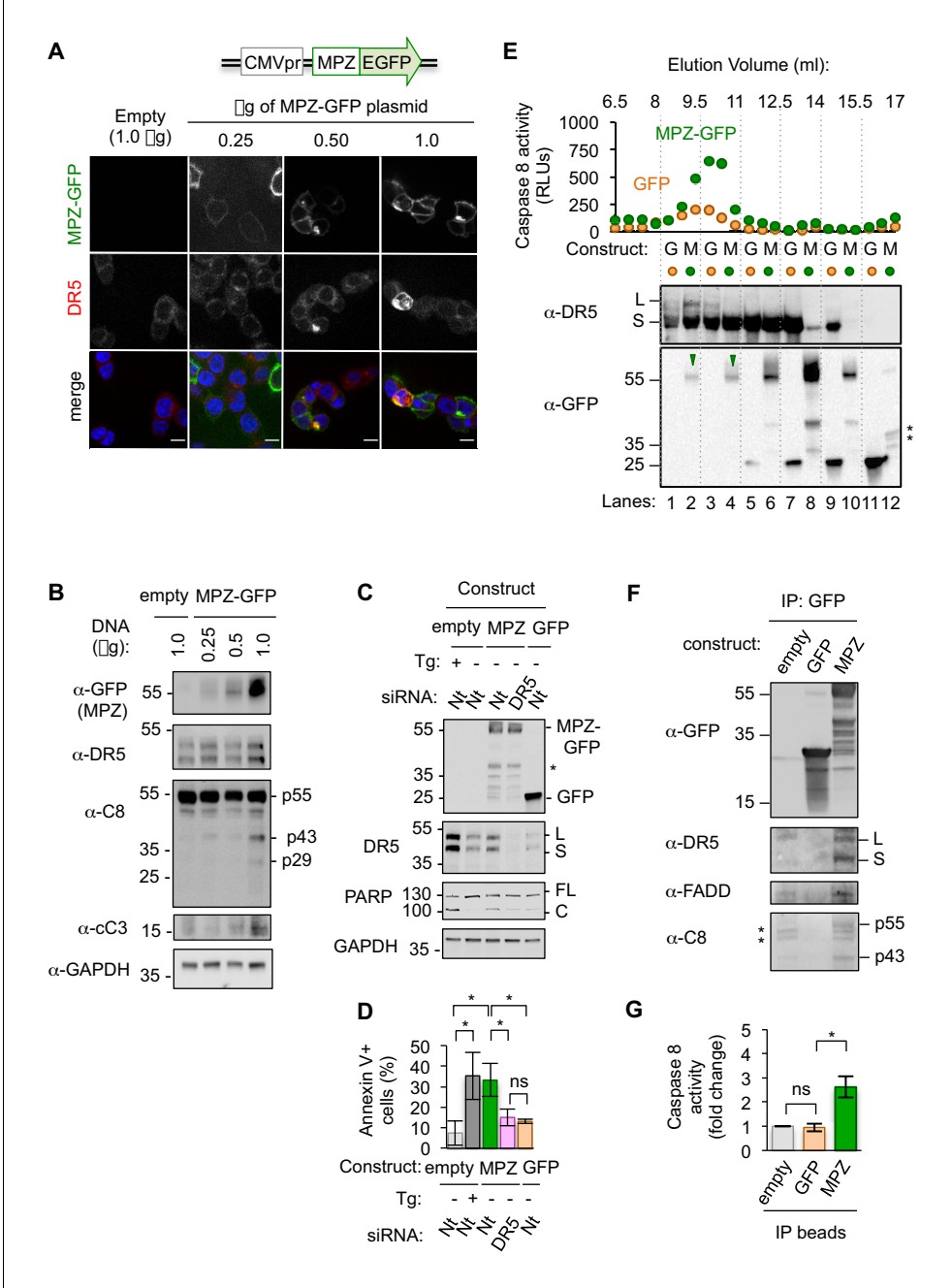

**Figure 1.** Misfolded proteins induce DR5-dependent apoptosis and assemble DR5-caspase 8 signaling complexes. (**A**) Confocal images of epithelial cells HCT116 fixed 24 hr post-transfection with 0.25–1.0 μg of a plasmid containing myelin protein zero (MPZ) tagged with a C-terminal monomeric EGFP or 1.0 μg of the empty vector showing MPZ-GFP fluorescence (green) and immunofluorescence with an antibody against DR5 (red) (scale bar = 5 μm). (**B**) Western blot of HCT116 cell lysates harvested 24 hr post-transfection with a titration of MPZ-GFP plasmid or the empty vector (C8 = caspase 8, cC3 = cleaved caspase 3). p55 represents full-length, inactive C8; p43 indicates a C8 intermediate after release of the active p10 subunit, and p29 corresponds to the released p18 and p10 subunits. (**C**) Western blot of HCT116 cells transfected with siRNA against a non-targeting (Nt) control or DR5 (48 hr) followed by the empty vector -/+ 100 nM thapsigargin (Tg), 1.0 μg MPZ-GFP, or cytosolic GFP (24 hr; * denotes degradation products; L and S denote the long and short isoforms of DR5, respectively; FL and C denote full-length and cleaved PARP, respectively). (**D**) Average percent of annexin V staining for HCT116 cells transfected as described in C) from n = 3 biological replicates (error bars = SEM; * indicates p<0.05; ns indicates p=0.46 as analyzed by unpaired t-test with equal SD). See ***Figure 1—figure supplement 4D*** for gating. (**E**) Top: Caspase 8

*Figure 1 continued on next page*

*Figure 1 continued*

activity in size exclusion chromatography fractions from lysates of HCT116 cells transfected with MPZ-GFP or cytosolic GFP (24 hr). Bottom: Size exclusion fractions were pooled according to dotted grid lines and immunoblotted for DR5 and GFP (* denotes degradation products). (F) Immunoprecipitation of GFP-tagged proteins from lysates of HCT116 transfected with MPZ-GFP, cytosolic GFP, or the empty vector (L and S denote the long and short isoforms of DR5, respectively). The percent of total DR5 recovered has been quantified in *Figure 1—figure supplement 5C*. (G) Fold change in caspase 8 activity relative to the empty vector control for beads with immunoprecipitated contents shown in *Figure 1F* (error bars = SEM for n = 3 biological replicates; * indicates p=0.023 and ns indicates p=0.83 as calculated by unpaired t-tests with equal SD).

The online version of this article includes the following source data and figure supplement(s) for figure 1:

**Source data 1.** Caspase glo 8 measurements for IP of MPZ-GFP vs GFP.
**Source data 2.** Westerns and quantification of DR5 recovered on IPs.
**Source data 3.** FCS files and quantification of annexin V staining for MPZ-GFP.
**Source data 4.** qPCR analysis of MPZ-GFP titration.
**Source data 5.** Caspase glo 8 measurements for time course of MPZ-GFP transfection.
**Source data 6.** qPCR and cell death measurement for CHOP expression.
**Source data 7.** qPCR analysis of INS and RHO-GFP expression.
**Source data 8.** FCS files and quantification of annexin V staining for INS and RHO.
**Source data 9.** Caspase glo 8 measurements for IP of INS and RHO-GFP.
**Figure supplement 1.** Sustained MPZ-GFP expression invokes a terminal, pro-apoptotic UPR at late time points.
**Figure supplement 2.** Upregulating DR5 levels in the absence of ER stress through ectopic expression of CHOP is not sufficient to induce apoptosis.
**Figure supplement 3.** DR5 immunoprecipitates with FADD and MPZ-GFP.
**Figure supplement 4.** Sustained overexpression of other ER-trafficked proteins induce UPR-mediated apoptosis in a DR5-dependent manner.
**Figure supplement 5.** DR5 engages a selective subset of ER-trafficked client proteins upon prolonged ER stress.

---

proinsulin (INS), which are also associated with CHOP-dependent cell death pathologies (*Chiang et al., 2016*; *Oyadomari et al., 2002*). Sustained overexpression of both RHO-GFP and INS-GFP upregulated BiP and CHOP mRNAs (*Figure 1—figure supplement 4A*) and induced *XBP1* mRNA splicing (*Figure 1—figure supplement 4B*). Both proteins formed SDS-insoluble aggregates and induced PARP cleavage and annexin V staining in a DR5-dependent manner (*Figure 1—figure supplement 4C–4E*). By contrast, immunoprecipitation of RHO-GFP enriched for DR5 protein and caspase 8 activity more robustly than INS-GFP (*Figure 1—figure supplement 5*), despite inducing DR5-dependent apoptosis to a similar extent. This indicates that misfolded proteins differ in their propensity to directly engage the DR5-assembled DISC, and that other misfolded substrates—caused by the ectopically overexpressed ER-trafficked protein—may mediate direct DR5 activation. Thus, as exemplified by MPZ and RHO, a selective subset of misfolded proteins in the secretory pathway can engage DR5 to form oligomeric complexes that induce caspase 8 activation.

## Misfolded protein engages DR5 at the ER-Golgi intermediate compartment, inducing active DR5 signaling clusters

To explore where within in the cell DR5 associated with misfolded protein, we used confocal imaging of fixed cells for immunofluorescence. These analyses revealed that intracellular MPZ-GFP and DR5 appeared in discrete puncta that often overlapped (*Figure 2A*). DR5 siRNA knockdown eliminated the DR5 signal, confirming the specificity of the DR5 antibody (*Figure 2—figure supplement 1A*, right panel). Similarly, overexpression of RHO also resulted in intracellular puncta that frequently co-localized with DR5 clusters (*Figure 2—figure supplement 1B*). Quantification of the mean Pearson's correlation per cell demonstrated statistically significant overlap with DR5 signal for both GFP-tagged MPZ and RHO (*Figure 2—figure supplement 1C*), indicating that these misfolded proteins accumulate in the same compartment as DR5.

Previous findings suggested that DR5 is retained near the Golgi apparatus during ER stress (*Lu et al., 2014*). We confirmed co-localization with the purported Golgi marker RCAS1, as previously reported (*Figure 2—figure supplement 1D*). However, we observed little overlap in DR5 staining with another *cis*-Golgi marker, giantin (*Figure 2E*). To resolve this discrepancy, we employed subcellular fractionation as an orthogonal biochemical approach. Separating organelle membranes

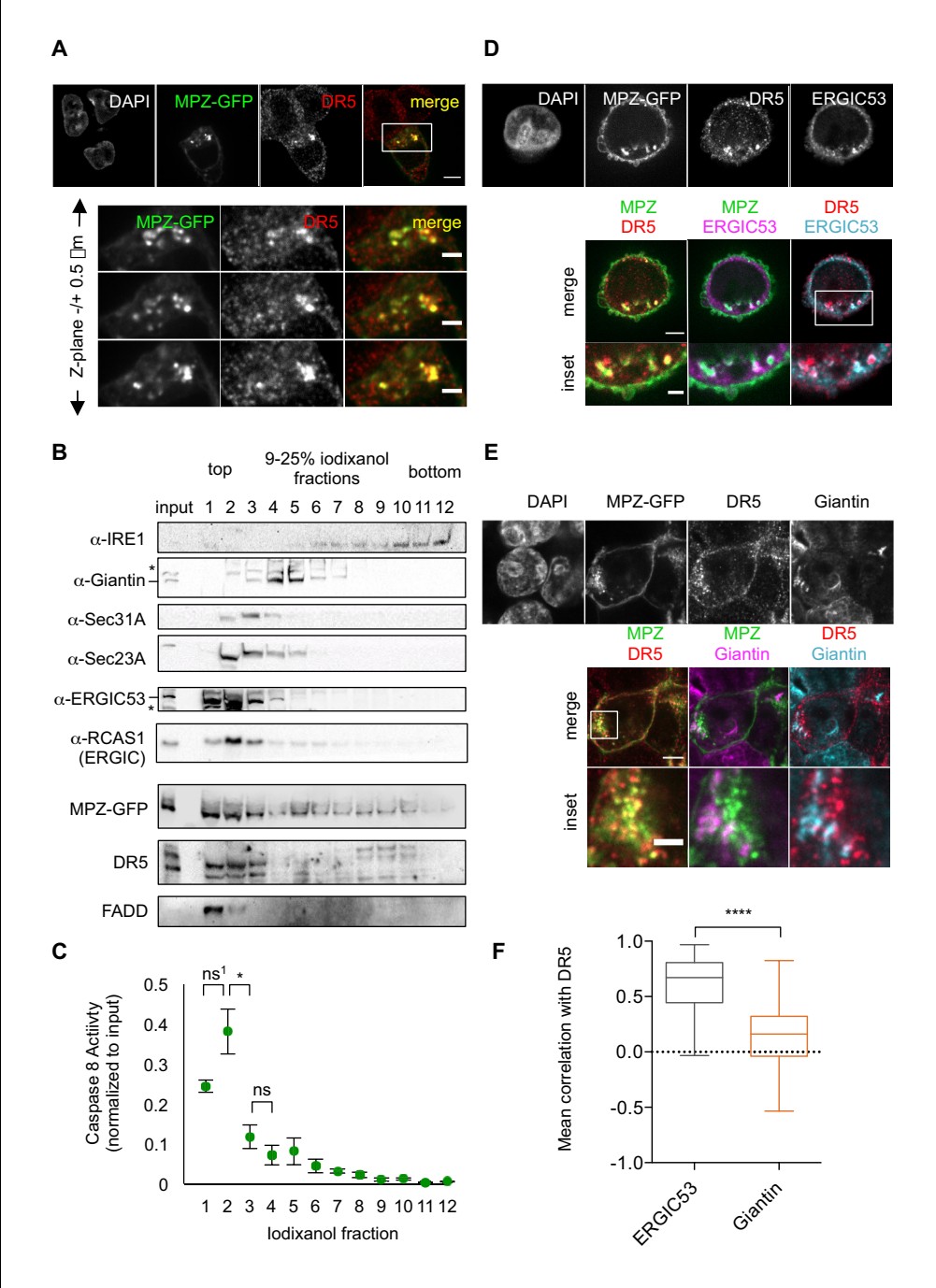

**Figure 2.** Misfolded protein engages DR5 at the ER-Golgi intermediate compartment, inducing active DR5 signaling clusters. (**A**) Top: Immunofluorescence of HCT116 cells transfected with MPZ-GFP (green) for 24 hr and stained with anti-DR5 (red, scale bar 5 μm). Bottom: Enlargements of the inset stepping through the z-plane in 0.5 μm increments (scale bar 2 μm). (**B**) Subcellular fractionation of lysate expressing MPZ-GFP, where IRE1 marks the ER, Giantin marks the Golgi, Sec31A and Sec23A mark COPII vesicles, and ERGIC53 and RCAS1 correspond to ER-Golgi intermediate compartment. Bands of the expected size are indicated by "– "and bands that may represent a modified or degraded protein are indicated by *. (**C**) Average caspase activity of each fraction from subcellular gradient centrifugation in (**B**) normalized to total lysate (input) measured by caspase 8 substrate luminescence (n = 3 biological replicates, error bars = SEM; ns[1] indicates p=0.079, * denotes p=0.015, and ns indicates p=0.31 from unpaired t-tests with equal SD). (**D**) Top: Immunostaining of DR5 and ERGIC53 in fixed HCT116 cells transfected with MPZ-GFP for 24 hr as in (**A**). Bottom: Merged images with ERGIC53 in magenta or cyan to depict overlapping signal as white (scale bar = 5 μm, insets scale bar = 2 μm). (**E**) Immunostaining of DR5 and giantin in

*Figure 2 continued on next page*

*Figure 2 continued*

fixed HCT116 cells expressing MPZ-GFP. Giantin is magenta in the overlay with MPZ (green) or cyan in the overlay with DR5 (red). Bottom row enlarges the inset marked in the merges images to show little overlapping signal with giantin (scale bar = 5 μm, inset scale bar = 1 μm). (F) Box-whisker plots quantifying the Pearson's correlation per cell between DR5 and ERGIC53 (mean = 0.61 ± 0.03) or giantin (mean = 0.14 ± 0.02) within MPZ-positive cells (N > 55), where whiskers correspond to minimum and maximum values of the data (**** indicates p<0.001). The online version of this article includes the following source data and figure supplement(s) for figure 2:

**Source data 1.** Caspase activity for fractions of iodixanol gradient.
**Figure supplement 1.** Intracellular puncta of overexpressed MPZ and rhodopsin proteins show significant co-localization with DR5 clusters.
**Figure supplement 2.** Misfolded protein accumulation in the ERGIC precedes DR5 retention in the ERGIC.

---

revealed that RCAS1, DR5, and MPZ-GFP co-sedimented in fractions containing ERGIC53, a marker of the ER-Golgi intermediate compartment (ERGIC), but not with those containing giantin (*Figure 2B*). Notably, a portion of FADD, a cytosolic protein expected to exclusively remain in the topmost, cytosolic fraction, migrated into the second fraction of the gradient, indicating its association with the ERGIC membranes. Consistent with the presence of FADD, the first and second ERGIC-associated fractions harbored the majority of the caspase 8 activity in the cell lysate (*Figure 2C*), indicating the presence of active DR5 DISCs. Moreover, immunofluorescence with quantification of the mean correlation per cell demonstrated the co-localization of DR5 with the ERGIC rather than with the Golgi (*Figure 2D and F*).

To determine when DR5 accumulates at the ERGIC relative to misfolded proteins, we compared the immunofluorescence of cells fixed at 20 hr (before the onset of caspase activity) and at 24 hr post-transfection (after the onset of caspase activity, *Figure 1—figure supplement 1E*). Intracellular puncta of MPZ appeared at 20 hr, preceding the appearance of DR5 clusters at 24 hr (*Figure 2—figure supplement 2A*). Between 20 and 24 hr, the correlation of DR5 and ERGIC53 increased, whereas the correlation of MPZ with ERGIC53 remained steady, indicating that DR5 accumulated after saturation of MPZ levels at the ERGIC (*Figure 2—figure supplement 2B–2C*). By contrast, the mean Pearson's correlation with giantin approached zero for both MPZ and DR5 at 24 hr post-transfection (*Figure 2—figure supplement 2B*, *Figure 2F*). These results confirm the localization of DR5 and misfolded protein at the ERGIC under conditions of unmitigated ER stress.

## Polypeptide sequences of mammalian ER-trafficked protein directly bind to the DR5 ectodomain and induce its oligomerization

With evidence of a physical association between misfolded protein and active DR5 oligomers at the ERGIC, we asked how misfolded proteins and DR5 interact. Considering the precedence that (i) DR5 binds unstructured peptides mimicking TRAIL (*Kajiwara, 2004*; *Pavet et al., 2010*) and (ii) that UPR sensors can directly bind misfolded protein to sense ER stress (*Karagöz et al., 2017*; *Wang et al., 2018*; *Gardner and Walter, 2011*), we hypothesized that DR5 may directly recognize unstructured regions of misfolded proteins through its ectodomain (ECD) that would project into the ERGIC lumen. Probing a peptide array with purified recombinant Fc-tagged DR5 ECD revealed promiscuous recognition of amino acid sequences throughout the ectodomain of MPZ and within extracellular loops of RHO (*Figure 3A*, *Figure 3—figure supplement 1A–1B*). Quantification of the relative signal intensity revealed that DR5-binding sequences were enriched for aliphatic and aromatic residues whereas polar and acidic residues were excluded (*Figure 3—figure supplement 1C*), reminiscent of qualities that become surface-exposed in misfolded or unfolded proteins.

To validate the specificity of DR5 interactions on the array, we performed pull-down assays on the MPZ-derived peptide exhibiting the strongest signal (spots C18-C19 in *Figure 3A*, hereon referred to as MPZ-ecto) with recombinant Fc-tagged DR5 ECD versus TNFR1 ECD as a selectivity control. The MPZ-ecto peptide bound specifically to the DR5 ECD but not the TNFR1 ECD (*Figure 3B*). Under equilibrium conditions, interaction with MPZ-ecto peptide quenched fluorescently labeled DR5 ECD but not fluorescently labeled TNFR1 ECD, yielding an apparent binding affinity of $K_{1/2}$ = 109 μM±11 μM with a Hill coefficient of 2.6 (*Figure 3C*, *Figure 3—figure supplement 2A*). Adding excess unlabeled DR5 ECD restored fluorescence (*Figure 3—figure supplement 2B*), indicating that the quenching reflected a specific and reversible interaction between the DR5

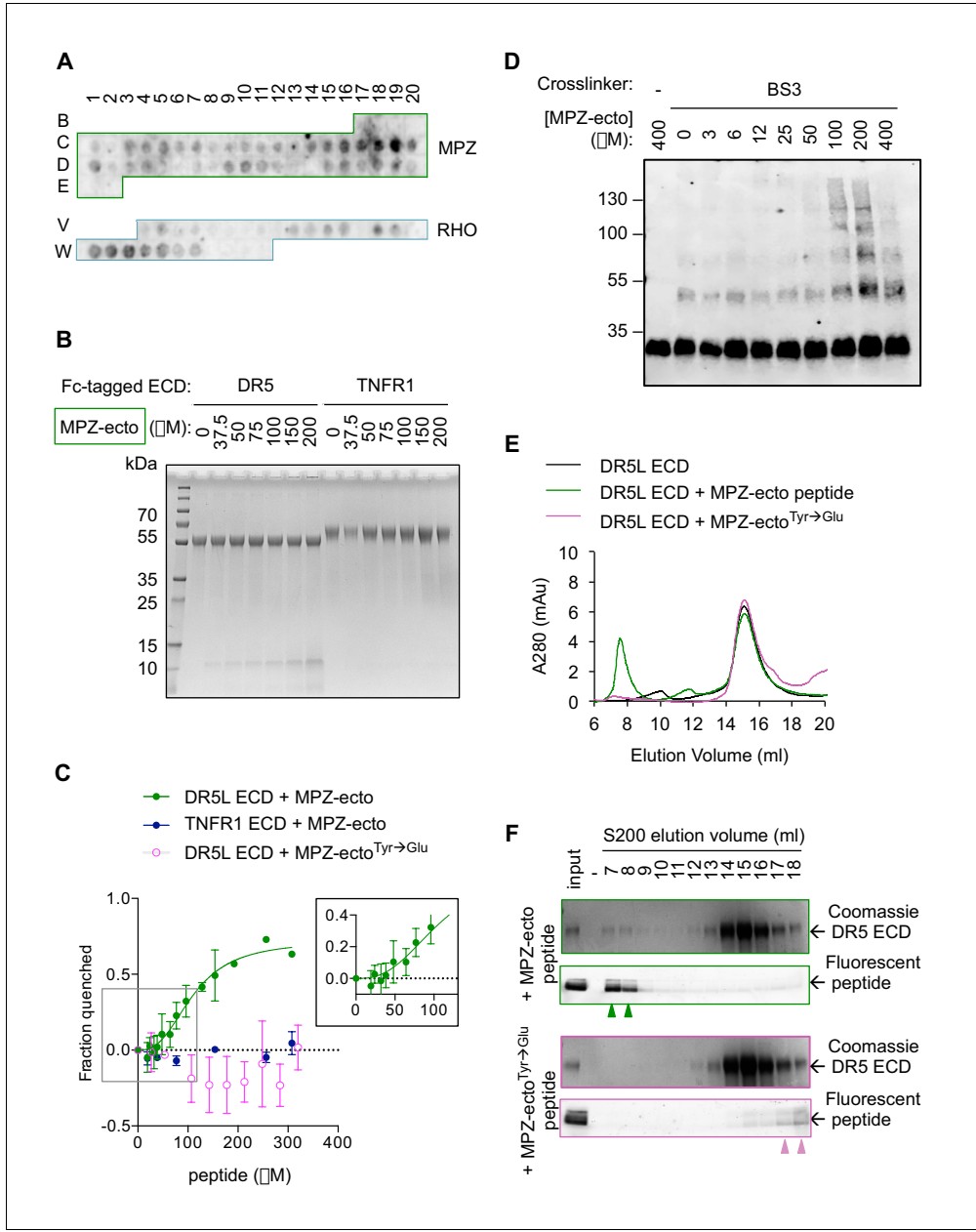

**Figure 3.** Direct binding of exposed ER-trafficked protein sequences to the DR5 ECD is sufficient to induce oligomerization. (**A**) A peptide array tiled with sequences from the ectodomain of myelin protein zero (MPZ) and extracellular loops from rhodopsin (RHO) was incubated with Fc-tagged DR5 ectodomain domain (long isoform, 500 nM). Signal was obtained by probing with anti-Fc. (**B**) Coomassie stained SDS-PAGE gel of pulldown on Fc-tagged DR5L ECD (55 kDa) or TNFR1 ECD (65 kDa) incubated with increasing concentrations of the MPZ-ecto$^{VD}$ peptide (apparent MW of 10 kDa, see 'Amino acid sequences of MPZ-derived peptides' for sequence). (**C**) Fluorescence quenching of AlexaFluor647-DR5L (green) or TNFR1 ECD (blue) was measured with increasing MPZ-ecto peptide to quantify the binding affinity, whereas quenching was not observed with the mutated MPZ-ecto$^{Tyr\rightarrow Glu}$ peptide (magenta) (N = 3, error bars are SD). DR5L ECD binds to the MPZ-ecto peptide with a $K_{1/2}$ of 109 ± 11 µM with a hill coefficient of 2.6 ± 0.5. (**D**) SDS-PAGE of recombinant FLAG-tagged DR5L ECD (25 kDa, 10 µM) incubated with MPZ-ecto peptide at the noted concentrations and treated with the amine crosslinker BS3 (100 µM), probed with anti-FLAG. (**E**) Size exclusion chromatographs of absorbance at 280 nm for 25 µM recombinant DR5L ECD alone (black), pre-incubated with 100 µM fluorescein-conjugated MPZ-ecto peptide (green) or 100 µM fluorescein-conjugated MPZ-ecto$^{Tyr\rightarrow Glu}$ peptide (magenta). (**F**) SDS-PAGE gels scanned for fluorescence and then stained with Coomassie for eluted size exclusion fractions in (**E**). Green outlines (top pair) correspond to fractions from DR5L pre-incubated with MPZ-ecto peptide, and magenta outlines (bottom pair) correspond to DR5L with

*Figure 3 continued on next page*

*Figure 3 continued*

MPZ-ecto$^{Tyr \to Glu}$ peptide. Lane marked by "-" denotes a blank lane between the input and 7 ml fraction to minimize spillover of signal from input sample. Arrowheads mark detectable peptide fluorescence in the indicated fractions.

The online version of this article includes the following source data and figure supplement(s) for figure 3:

**Source data 1.** Sequences and quantification of peptides probed with Fc-DR5 ECD on the peptide array.
**Figure supplement 1.** DR5 ECD binds to selective subset of sequences displayed by the secretory proteome.
**Figure supplement 2.** Purified recombinant DR5 ECD oligomerizes with peptide in a specific and reversible manner.

---

ECD and the MPZ-ecto peptide. Moreover, mutation of two aromatic amino acids (both Tyr) to disfavored acidic amino acids (Glu) abrogated binding (*Figure 3C*), demonstrating that the interaction is sequence-specific.

The Hill coefficient of 2.6 suggested cooperative binding. Therefore, we tested if the DR5 ECD forms oligomers in the presence of peptide. In the absence of peptide, the addition of a chemical cross-linker captured dimers of FLAG-tagged DR5 ECD (*Figure 3D*, *Figure 3—figure supplement 2C*), consistent with pre-ligand assembled dimers previously observed for members of the TNFR family (*Clancy et al., 2005*; *Siegel et al., 2000*; *Chan et al., 2000*). With increasing concentration of peptide (up to 200 µM), crosslinking revealed multimers of the DR5 ECD (*Figure 3D*), indicating that the peptide acts as a ligand to template assembly of DR5 oligomers. Interestingly, excess peptide (400 µM) dissociated higher-order oligomers of DR5, suggesting a lower valency of interaction when the DR5 concentration becomes limiting.

To examine the DR5 oligomerization at saturating peptide concentrations by an orthogonal method, we fractionated DR5 ECD-peptide complexes using size exclusion chromatography. At 100 µM MPZ-ecto (~$K_{1/2}$), DR5 ECD co-eluted with the peptide as higher-order oligomers near the void volume (7–8 ml) and as apo-dimers centered at 14 ml, as shown in the Coomassie blue-stained gel for DR5 and fluorescence scan for fluorescein-labeled MPZ-ecto peptide (*Figure 3E, F*, green outline). This elution pattern was similar to that of the DR5 ECD-TRAIL complex, for which both proteins co-eluted near the void volume (*Figure 3—figure supplement 2E–2F*). However, with excess MPZ-ecto peptide at 400 µM (4-times $K_{1/2}$), the proportion of higher-order oligomers of DR5 ECD and the peptide diminished and re-distributed to later eluting fractions at 12–15 ml (*Figure 3—figure supplement 2G–2H*, teal outline), indicating disassembly into smaller oligomers of DR5 ECD and pointing at the reversibility of the higher-order DR5-peptide assemblies. Importantly, the non-binding peptide bearing the Tyr-to-Glu substitutions did not co-migrate with or induce the oligomerization of DR5 ECD (*Figure 3E, F*, magenta outline).

## Disrupting misfolded protein binding to DR5 attenuates ER stress-mediated apoptosis

Since mutating the Tyr residues to Glu on the MPZ-ecto peptide proved sufficient to disrupt the DR5 ECD interaction in solution, we tested the ability of this minimal MPZ-derived sequence to bind to and activate DR5 in cells. To this end, we generated constructs that replaced the ectodomain of MPZ with either the MPZ-ecto peptide, the peptide sequence with Tyr-to-Glu substitutions, or the peptide with all its aromatic residues changed to Glu to further deplete DR5-favored amino acid side chains revealed in the peptide array (*Figure 4A*). In a titration of MPZ-ecto peptide expression, the WT peptide sequence induced more PARP cleavage and caspase activity than similar or higher levels of the peptides containing Glu substitutions (*Figure 4B*, compare lanes 5, 7, and 10, *Figure 4C*). The Glu-containing peptides also induced reduced PARP cleavage in another epithelial cell type, HepG2 (*Figure 4—figure supplement 1A*). Acute knockdown of DR5 reduced PARP cleavage during MPZ-ecto peptide expression, while exogenous FLAG-tagged DR5 expression restored PARP cleavage (*Figure 4—figure supplement 1B*). Of note, depletion of DR5 resulted in detection of higher levels of the MPZ-ecto peptide, likely because cells with this protein-folding burden were not eliminated.

Expressing comparable levels of the MPZ-ecto peptide and its variants (using conditions of lanes 5, 7, and 10 in *Figure 4B*) induced *XBP1* mRNA splicing and transcription of CHOP and BiP mRNAs,

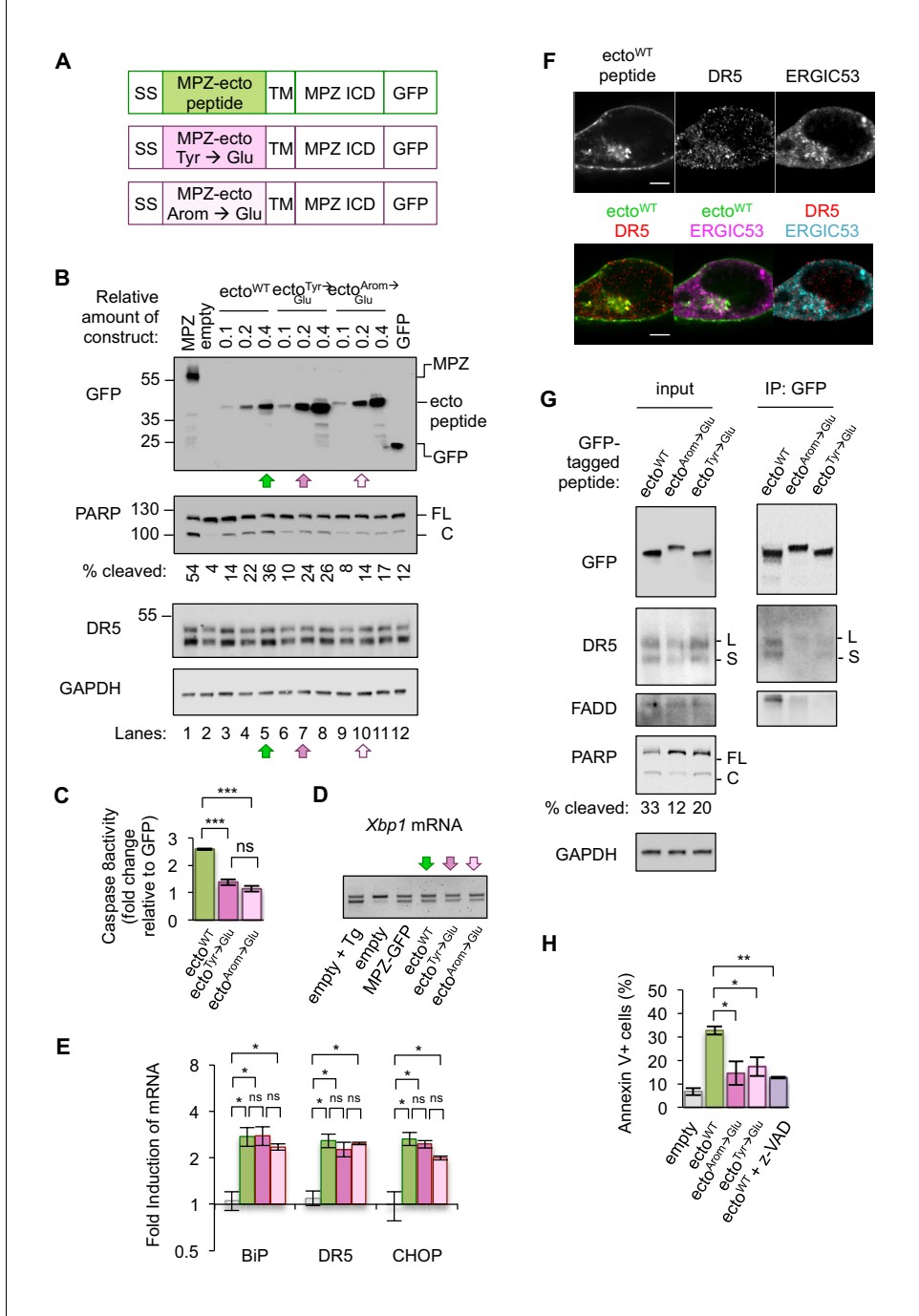

**Figure 4.** Disrupting misfolded protein binding to DR5 impairs ER stress-induced apoptosis. (**A**) Diagram of constructs generated to replace the MPZ ectodomain with the minimal DR5-binding MPZ-ecto peptide (green), the peptide harboring Tyr → Glu mutations (magenta), or the peptide with all aromatic residues (Arom) mutated to Glu (light pink). SS = signal sequence of MPZ, TM = transmembrane domain, ICD = intracellular domain. (**B**) Western blot of HCT116 cell lysates harvested 24 hr post-transfection with 1 μg of MPZ-GFP plasmid, empty vector, or a titration of GFP-tagged MPZ-ecto peptide variants, followed by GFP alone. FL denotes full-length PARP, while C denotes cleaved PARP. The percentage of cleaved PARP was calculated as the signal of cleaved PARP divided by total PARP (FL + C). Arrows denote conditions carried forward for normalized expression levels of the ecto peptide constructs. (**C**) Fold change in caspase8 activity relative to GFP expression, as measured by incubation of luminescent caspase glo 8 substrate with lysates from HCT116 transfected using conditions described in *Figure 4B* lanes 5, 7 and 10 (error bars represent SEM of n = 3 biological replicates; *** denotes p<0.005, and ns indicates p=0.18 from unpaired t-tests with equal SD). (**D**) RT-PCR for unspliced and spliced forms

*Figure 4 continued on next page*

*Figure 4 continued*

of *Xbp1* mRNA isolated from HCT116 cells transfected for 24 hr with the empty vector + / - 100 nM Tg, or with MPZ-GFP, or MPZ-ecto peptide-GFP and its mutant variants (Tyr → Glu and Arom → Glu) using conditions from *Figure 4B*, lanes 5, 7, and 10. (E) qPCR for reverse-transcribed transcripts harvested from HCT116 cells transfected with the constructs described in 4A, using conditions shown in *Figure 4B* lanes 5, 7, and 10. (n = 3 biological replicates, * denotes p<0.05 and ns = non significant). (F) Immunofluorescence for DR5 and ERGIC53 in HCT116 transfected with the MPZ-ecto peptide for 24 hr. ERGIC53 is magenta in the overlay with MPZ (green) or cyan in the overlay with DR5 (red) (scale bar = 5 μm). (G) Left: Immunoblots of HCT116 lysate inputs expressing the constructs described in 4A, where L and S mark the long and short isoforms of DR5, respectively, and where FL and C mark the full-length and cleaved fragments of PARP, respectively. The percentage of cleaved PARP is quantified as the signal of the cleaved fragment divided by total PARP (FL + C). Right: Immunoprecipitation of GFP-tagged proteins from the lysates shown in (C), where L and S denote the long and short isoforms of DR5, respectively. (H) Average percent of annexin V staining for HCT116 cells transfected as described in C) and D) from n = 3 biological replicates (error bars = SEM, * indicates p=0.026, and ** indicates p=0.003 from unpaired t-tests with equal SD). See *Figure 4—figure supplement 3* for distribution of early vs late apoptotic cells.

The online version of this article includes the following source data and figure supplement(s) for figure 4:

**Source data 1.** Westerns and quantification of DR5 recovered on IPs.
**Source data 2.** Caspase glo 8 measurements for MPZ-ecto peptide expression.
**Source data 3.** qPCR and statistical analysis for expression of MPZ-ecto peptides.
**Source data 4.** FCS files and quantification of annexin V staining for MPZ-ecto peptides.
**Figure supplement 1.** Introducing Glu mutations to the DR5-binding sequence of MPZ disrupts PARP cleavage in a DR5-dependent manner.
**Figure supplement 2.** Glu-containing mutants of MPZ-ecto peptide accumulate in the ERGIC.
**Figure supplement 3.** MPZ-ecto peptide engagement of DR5 in cells drives apoptotic cell death.

---

indicating that the presence of these peptides perturb ER protein folding homeostasis to a similar degree (*Figure 4D*, *Figure 4E*). Immunofluorescence showed that the MPZ-ecto peptide localized to the plasma membrane and within intracellular puncta that partially overlapped with ERGIC signal, although to a lesser extent than overexpressed full-length MPZ (*Figure 4F*, *Figure 4—figure supplement 2C*). The Glu-containing mutant peptides were similarly distributed within cells with no significant difference in their average correlation with ERGIC signal (*Figure 4—figure supplement 2A–2B*). DR5, in all three conditions, also showed a positive correlation with the ERGIC marker (*Figure 4—figure supplement 2E*). To determine if DR5 interacted with the MPZ-ecto peptide or its mutants, we immunoprecipitated the GFP-tagged peptides. Pulldown of the MPZ-ecto peptide enriched for DR5 relative to the Glu-containing mutant peptides (*Figure 4G*, *Figure 4—figure supplement 3A*). Consistent with this specific enrichment of DR5 for the WT sequence, PARP cleavage and caspase activity measured in cell lysates were increased with the WT MPZ-ecto relative to the mutants (*Figure 4B,C*). To confirm that the expression of MPZ-ecto peptide induces apoptotic cell death, we measured annexin V staining in the absence and presence of the pan-caspase inhibitor z-VAD (*Figure 4H*, *Figure 4—figure supplement 2C–2D*). As expected, expressing the MPZ-ecto peptide increased annexin V staining relative to the empty vector but treatment with zVAD diminished the extent of annexin V staining (*Figure 4H*). Importantly, cells expressing the Glu-containing mutant peptides exhibit decreased annexin V staining, demonstrating that DR5 binding of exposed polypeptides on misfolded protein is important for driving apoptosis.

## Discussion

Our data identify misfolded protein as the ER stress factor that switches upregulated DR5 from its inactive auto-inhibited dimer state to active multimeric clusters to initiate DISC assembly and apoptosis at the ER-Golgi intermediate complex. We have examined the mechanism of apoptosis induction by the sustained expression of three different candidate ER-trafficked proteins associated with CHOP-dependent disease pathologies: MPZ, RHO, and INS (*Pennuto et al., 2008*; *Chiang et al., 2016*; *Oyadomari et al., 2002*). In epithelial cells, overexpression of each protein induces apoptosis in a DR5-dependent manner. Consistent with previous reports of ectopic CHOP expression in the absence of ER stress (*McCullough et al., 2001*; *Han et al., 2013*; *Southwood et al., 2016*), CHOP-driven upregulation of DR5 alone did not account for the apoptosis observed during the

overexpression of an ER-trafficked protein. For MPZ and RHO, the intracellular, misfolded pools of each protein physically associated with the DR5-caspase 8 complex. For proinsulin, which weakly associated with DR5 but triggered apoptosis to a similar extent, we believe it is likely that overexpression of this singular protein perturbed the folding of endogenous trafficking substrates and thereby provided other, perhaps more favored, misfolding substrates to directly engage DR5. This latter scenario is likely to occur under pharmacologically induced ER stress as well. The interaction between misfolded protein and DR5 bridges the long-standing mechanistic gap of why CHOP expression (and subsequent upregulation of its downstream factors) is necessary but not sufficient to drive cell death. Through characterizing the interaction between the DR5 ECD and peptide sequences of ER-trafficked proteins, we demonstrate that DR5 promiscuously binds to exposed hydrophobic stretches of misfolded proteins with an affinity in the range of 100 µM and in a highly cooperative manner.

To grasp how such a high concentration of misfolded protein could occur in the ERGIC, it is important to consider that the compartment is composed of vesicles and tubules measuring 60–100 nm in diameter and <500 nm in length. In a back-of-the-envelope calculation, we estimate that reaching 100 µM in a vesicle with a diameter of 100 nm would require only 32 molecules (*Sesso et al., 1994*; *Fan et al., 2003*). The measured affinities are therefore well within physiological range. Quantitative fluorescence microscopy of living COS7 cells has indicated up to 100 molecules of a GFP-tagged viral glycoprotein in a 100 nm vesicle (*Hirschberg et al., 1998*), providing experimental evidence that surpassing concentrations of 100 µM is indeed physiologically relevant. In fact, the 'low' affinity between DR5 and misfolded proteins is likely a necessary feature that prevents aberrant DR5 oligomerization and activation in the crowded lumenal environment of membrane-bound compartments, as we previously established for other unfolded protein sensors, such as IRE1 (*Gardner et al., 2013*; *Gardner and Walter, 2011*; *Karagöz et al., 2017*).

Given that misfolded receptors can be exported from the ER when quality control mechanisms are overwhelmed (*Satpute-Krishnan et al., 2014*; *Sirkis et al., 2017*), detection of misfolded proteins by DR5 downstream of the ER likely serves to prevent the cell from displaying or secreting dysfunctional proteins that would be detrimental in a multicellular context. While IRE1 and PERK act as initial UPR sensors in the ER, DR5 acts as a late sensor of misfolded protein at the ERGIC during unmitigated ER stress. Thus, intracellular DR5 triggers apoptosis to enforce a terminal quality control checkpoint for secretory and transmembrane proteins. We postulate that other members of the TNFR family, for exmple DR4, which has been reported to play a role in cell death during Golgi stress (*van Raam et al., 2017*), may respond similarly to intracellular stimuli.

Although extensive research has focused on the therapeutic activation of death receptors including DR5 (*Ashkenazi, 2015*), limited strategies exist to inhibit such receptors despite their demonstrated role in apoptosis-mediated disease progression (*Vunnam et al., 2017*). Namely, DR5-mediated apoptosis in hepatocytes has been linked to non-alcoholic fatty liver disease, while CHOP-dependent apoptosis in Schwann cells—wherein a role for DR5 has yet to be investigated—may contribute to diabetic peripheral neuropathies (*Cazanave et al., 2011*; *Sato et al., 2015*). Our finding that the assembly and disassembly of DR5 ECD oligomers can be controlled by a peptide raises the possibility that intracellular DR5 activation could be inhibited through small molecule ligand-induced dissociation of DR5 clusters to prevent apoptosis and thus preserve cell viability in the face of unresolved ER stress. From the work herein, this notion now emerges as a promising strategy to interfere therapeutically with deleterious death receptor function.

# Materials and methods

**Key resources table**

| Reagent type (species) or resource | Designation | Source or reference | Identifiers | Additional information |
|---|---|---|---|---|
| Gene (*Homo- sapiens*) | DR5 | | O14763 (TR10B_HUMAN) | |
| Gene (*Homo- sapiens*) | INS (proinsulin) | | P01308 (INS_HUMAN) | |
| Gene (*Homo- sapiens*) | MPZ (myelin protein zero) | | P25189 (MYP0_HUMAN) | |

*Continued on next page*

*Continued*

| Reagent type (species) or resource | Designation | Source or reference | Identifiers | Additional information |
|---|---|---|---|---|
| Gene (*Homo- sapiens*) | RHO (rhodopsin) | | P08100 (OPSD_HUMAN) | |
| Cell line (*Homo-sapiens*) | HCT116 | ATCC | CCL-247 | |
| Cell line (*Homo-sapiens*) | HepG2 | ATCC | CRL-10741 | |
| Transfected construct (*Homo sapiens*) | CHOP (canonical isoform) | this paper | | expression of CHOP in absence of ER stress used in *Figure 1— figure supplement 2* |
| Transfected construct (*Homo sapiens*) | cytosolic GFP | this paper | | expression of GFP used in *Figure 1C-H* |
| Transfected construct (*Homo sapiens*) | DR5 long FLAG-His6x, WT with silent mutation | this paper | | expression of DR5 long isoform-FLAG, harbors silent nt mutations within signal sequence to evade siRNA (AAGACCCTTGTGCTCGTTGTC à AAaACaCTTGTGCTCGTTGTC) used in *Figure 4—figure supplement 1* |
| Transfected construct (*Homo sapiens*) | DR5siRNA-2 | Dharmacon | siRNA | AAG ACC CUU GUG CUC GUU GUC UU, knockdown in Fig S9B |
| Transfected construct (*Homo sapiens*) | empty vector (no GFP) | this paper | | empty vector used in *Figures 1* and *4B, E* |
| Transfected construct (*Homo sapiens*) | MPZ-ecto peptide-eGFP | this paper | | expression of MPZ-ecto peptide (FTWRYQPEGGRDAISIFHYA) used in *Figure 4* |
| Transfected construct (*Homo sapiens*) | MPZ-ecto peptide$^{Arom \to Glu}$-eGFP | this paper | | expression of MPZ-ectoArom ->Glu peptide (ETEREQPEGGRDAISIEHEA) used in *Figure 4* |
| Transfected construct (*Homo sapiens*) | MPZ-ecto peptide$^{Tyr \to Glu}$-eGFP | this paper | | expression of MPZ-ectoTyr->Glu peptide (FTWREQPEGGRDAISIFHEA) used in *Figure 4* |
| Transfected construct (*Homo sapiens*) | MPZ-eGFP | this paper | | expression of MPZ used in *Figures 1* and *2, 4E* |
| Transfected construct (*Homo sapiens*) | Nt siRNA | Dharmacon | siRNA | AAA CCU UGC CGA CGG UCU ACC UU |
| Transfected construct (*Homo sapiens*) | ON-TARGETplus Human TNFRSF10B 8795 siRNA | Dharmacon | L-004448-00-0005 | *Figure 1* knockdowns |
| Transfected construct (*Homo sapiens*) | ON-TARGETplus Non-targeting siRNA #2 | Dharmacon | D-001810-02-05 | *Figure 1* knockdowns |
| Transfected construct (*Homo sapiens*) | proinsulin (INS) -GFP | this paper | | expression of INS-GFP used in *Figure 1—figure supplements 4* and *5* |
| Transfected construct (*Homo sapiens*) | rhodopsin (RHO) -GFP | this paper | | expression of RHO-GFP used in *Figure 1—figure supplements 4* and *5*, *Figure 2—figure supplement 1* |
| Antibody | anti-caspase 3 (rabbit) | Cell Signaling Technology | 9662 | 1:1000 for Westerns |
| Antibody | anti-caspase 8 5F7 (mouse) | MBL International | M032-3 | 1:1000 for Westerns |
| Antibody | anti-cleaved caspase 8 (Asp391) (18C8) (rabbit) | Cell Signaling Technology 9496 | 9496 | 1:50 for IF fixed with 4% PFA |

*Continued on next page*

*Continued*

| Reagent type (species) or resource | Designation | Source or reference | Identifiers | Additional information |
|---|---|---|---|---|
| Antibody | anti-DR5 (rabbit) | Cell Signaling Technology | 8074 | 1:1000 for Westerns |
| Antibody | anti-DR5 3H3 (mouse) | Genentech | | 1:100 for IF fixed with 4% PFA |
| Antibody | anti-ERGIC53 (rabbit) | Sigma Aldrich | E1031 | 1:1000 for Westerns |
| Antibody | anti-ERGIC53 (rabbit) | Sigma Aldrich E1031 | E1031 | 1:100 for IF fixed with methanol |
| Antibody | anti-FADD (mouse) | BD Biosciences | 610400 | 1:1000 for Westerns |
| Antibody | anti-Fc (mouse) | One World Lab | #603–510 | 1:1000 for Westerns |
| Antibody | anti-FLAG M2 (mouse) | Sigma | F1804 | 1:1000 for Westerns |
| Antibody | anti-GAPDH (rabbit) | Abcam | 9485 | 1:1000 for Westerns |
| Antibody | anti-GFP (mouse) | Roche | 11814460001 | 1:1000 for Westerns |
| Antibody | anti-Giantin (rabbit) | Abcam | ab24586 | 1:1000 for Westerns |
| Antibody | anti-Giantin (rabbit) | Abcam ab24586 | ab24586 | 1:1000 for IF fixed with methanol |
| Antibody | anti-His 6x (mouse) | Abcam | ab15149 | 1:1000 for Westerns |
| Antibody | anti-IRE1 14C10 (rabbit) | Cell Signaling Technology | 3294 | 1:1000 for Westerns |
| Antibody | anti-mouse-AlexaFluor546 | Invitrogen | A11030 | 1:1000 for IF, centrifuged at 15000xg for 20 mins at 4°C to remove aggregates |
| Antibody | anti-PARP (rabbit) | Cell Signaling Technology | 9542 | 1:1000 for Westerns |
| Antibody | anti-rabbit-AlexaFluor633 | Invitrogen | A21071 | 1:1000 for IF, centrifuged at 15000xg for 20 mins at 4°C to remove aggregates |
| Antibody | anti-RCAS1 D2B6N XP (rabbit) | Cell Signaling Technology | 12290 | 1:1000 for Westerns |
| Antibody | anti-RCAS1 D2B6N XP (rabbit) | Cell Signaling Technology 12290 | 12290 | 1:200 for IF fixed with methanol |
| Antibody | anti-Sec23A (rabbit) | Invitrogen | PA5-28984 | 1:1000 for Westerns |
| Antibody | anti-Sec31A (mouse) | BD Biosciences | 612350 | 1:1000 for Westerns |
| Recombinant DNA reagent | Gp64-His6x-DR5 long ECD pFastBacHT | this paper | plasmid | recombinant protein expression in SF21, used in *Figure 3* |
| Recombinant DNA reagent | Gp64-His6x-DR5 long ECD-FLAG pFastBacHT | this paper | plasmid | recombinant protein expression in SF21, used in *Figure 3* |
| Recombinant DNA reagent | Gp64-His6x-TNFR1 ECD pFastBacHT | this paper | plasmid | recombinant protein expression in SF21, used in *Figure 3* |
| Sequence-based reagent | BiP (GRP78) | this paper | 5' primer | GTTCGTGGCGCCTTGTGAC |
| Sequence-based reagent | BiP (GRP78) | this paper | 3' primer | CATCTTGCCAGCCAGTTGGG |
| Sequence-based reagent | CHOP (DDIT3) | this paper | 5' primer | AGCCAAAATCAGAGCTGGAA |
| Sequence-based reagent | CHOP (DDIT3) | this paper | 3' primer | TGGATCAGTCTGGAAAAGCA |
| Sequence-based reagent | DR5 (TNFRSF10B) | this paper | 5' primer | TTCTGCTTGCGCTGCACCAGG |
| Sequence-based reagent | DR5 (TNFRSF10B) | this paper | 3' primer | GTGCGGCACTTCCGGCACAT |
| Sequence-based reagent | GADD34 | this paper | 5' primer | GAGGAGGCTGAAGACAGTGG |
| Sequence-based reagent | GADD34 | this paper | 3' primer | AATTGACTTCCCTGCCCTCT |
| Sequence-based reagent | GAPDH | this paper | 5' primer | AGCCACATCGCTCAGACAC |

*Continued on next page*

*Continued*

| Reagent type (species) or resource | Designation | Source or reference | Identifiers | Additional information |
|---|---|---|---|---|
| Sequence-based reagent | GAPDH | this paper | 3' primer | TGGAAGATGGTGATGGGATT |
| Sequence-based reagent | GFP | this paper | 5' primer | CTGACCTACGGCGTGC |
| Sequence-based reagent | GFP | this paper | 3' primer | CCTTGAAGAAGATGGTGCG |
| Sequence-based reagent | MPZ | this paper | 5' primer | GGCCATCGTGGTTTACAC |
| Sequence-based reagent | MPZ | this paper | 3' primer | GATGCGCTCTTTGAAGGTC |
| Sequence-based reagent | XBP1 splicing | | 5' primer | GGAGTTAAGACAGCGCTTGG |
| Sequence-based reagent | XBP1 splicing | | 3' primer | ACTGGGTCCAAGTTGTCCAG |
| Peptide, recombinant protein | Fc-tagged DR5 ECD | Genentech | recombinant protein | |
| Peptide, recombinant protein | Fc-tagged TNFR1 ECD | Genentech | recombinant protein | |
| Peptide, recombinant protein | MPZ-ecto | Genscript | purified peptide (>95%) | FTWRYQPEGGRDAISIFHYA |
| Peptide, recombinant protein | MPZ-ecto$^{Tyr \rightarrow Glu}$ | Genscript | purified peptide (>95%) | FTWREQPEGGRDAISIFHEA |
| Peptide, recombinant protein | MPZ-ecto$^{VD}$ | Genscript | purified peptide (>95%) | VSDDISFTWRYQPEGGRD |
| Chemical compound, drug | 32% paraformaldehyde | Electron Microscopy Sciences | | fixed with 4% PFA diluted directly into media |
| Chemical compound, drug | AlexaFluor647 NHS Ester (Succinimidyl Ester) | Life Technologies | A37573 | fluorescent protein labeling |
| Chemical compound, drug | Annexin V-AlexaFluor 647 conjugate | ThermoFisher | #A23204 | apoptosis assays |
| Chemical compound, drug | BS3 (bis(sulfosuccinimidyl) suberate) | ThermoFisher | 21580 | 100 µM BS3 for 20 min at RT |
| Chemical compound, drug | Cellfectin II | ThermoFisher | 10362100 | insect cell expression |
| Chemical compound, drug | Collagen IV | Sigma Aldrich | C6745 | 0.03 mg/ml in PBS incubated on glass slide for 30 min at RT, and then rinsed off with PBS x 4 |
| Chemical compound, drug | DAPI | Molecular Probes | D-1306 | 5 ug/ml |
| Chemical compound, drug | iQ SYBR Green Supermix | BioRad | #17088800 | qPCR |
| Chemical compound, drug | Lipofectamine-LTX | Life Technologies | #15338100 | |
| Chemical compound, drug | OptiPrep Density Medium | Sigma Aldrich | D1556 | iodixanol gradient |
| Commercial assay, kit | caspase glo 8 reagent | Promega | PRG8200 | |
| Software, algorithm | CellProfiler | https://cellprofiler.org/ | | quantification of mean correlation |
| Commercial assay, kit | Dynabeads Protein G | ThermoFisher | 10003D | |
| Software, algorithm | FlowJo | FlowJo, LLC | | |
| Commercial assay, kit | GFP-Trap magnetic agarose beads | Chromotek | gtma-20 | GFP pulldowns |
| Software, algorithm | Prism 6.0 | GraphPad | | Kd fits, statistical analyses |
| Other | 8-well glass bottom uSlide | Ibidi | 80827 | |

*Continued on next page*

*Continued*

| Reagent type (species) or resource | Designation | Source or reference | Identifiers | Additional information |
|---|---|---|---|---|
| Other | normal goat serum | Jackson Immunoresearch Laboratories | 005-000-121 | blocked with 2% goat serum diluted into PHEM buffer |
| Other | peptide array | MIT Biopolymers Laboratory | *Karagöz et al., 2017* | |
| Other | premium capillaries | Nanotemper Technologies | #MO-K025 | fluorescence quenching assays |
| Other | SuperDex200 10/300 GL | GE Healthcare | 28990944 | size exclusion column for fractionation |

## Cell culture and experimental reagents

HCT116 cells (ATCC CCL-247) and HepG2 cells (ATCC CRL-10741) were cultured in DMEM with high glucose (Sigma D5796) supplemented with 10% FBS (Life technologies # 10082147), 2 mM L-glutamine (Sigma G2150), 100 U penicillin, and 100 µg/mL streptomycin (Sigma P0781). Cells were incubated at 37°C, 5% CO2 for growth and transfections. All cell lines were authenticated by DNA fingerprint STR analysis by the American Type Culture Collection (ATCC). All cell lines were visually inspected using DAPI DNA staining and tested negative for mycoplasma. Thapsigargin was purchased from Sigma and used at 100 nM in 0.1% DMSO unless otherwise indicated.

## Statistical analyses

Unpaired two-tailed t-tests for data sets were performed using GraphPad Prism 6.0, where the variance between two data sets was non-significant, unless otherwise indicated.

## Transient transfections for protein expression

For each 6-well sample seeded with 400,000 cells the evening prior to transfection, the final transfection mixture put onto the cells was composed of 2 ml OptiMEM I (Thermo Fisher Scientific #31985070), 5 µl of Lipofectamine-LTX (Life Technologies #15338100), 1000 ng of total DNA (supplemented with the empty vector in cases of variable MPZ-GFP). Plasmid preparations were preformed fresh for each transfection to maximize reproducibility. Unless otherwise noted, 1.0 µg of plasmid containing GFP-tagged ER-trafficked protein was used, while 0.25 µg of GFP supplemented with 0.75 µg of empty vector was sufficient to yield GFP protein levels in excess of ER-trafficked GFP fusions.

To prepare the transfection mixture for one well of a 6-well plate, 5 µl of Lipofectamine-LTX and 1000 ng of plasmid were each diluted separately into 200 µl of OptiMEM I and then combined and incubated at RT for 15 min, as adapted from the manufacturer's protocol. Growth media for each 6-well sample was replaced with 1.5 ml of OptiMEM I and the 400 µl transfection mixture was added dropwise to each well and incubated for 24 hr (see Cell line culture conditions).

For transfections in 15 cm dishes or 8-well ibidi slides, the transfection mixture was scaled to the number of cells plated (i.e. 10 µg of plasmid for 4 million cells, or 62.5 ng of plasmid for 25,000 cells).

### Constructs used for protein expression in cells

| Lab archive | Plasmid description | Vector | Resistance | Purpose | Construct used in figure(s): |
|---|---|---|---|---|---|
| pPW3403 | empty (no GFP) | pEGFP | KanR | empty vector | *Figures 1, 4B, 4E* |
| pPW3404 | MPZ-eGFP | pEGFP | KanR | expression of MPZ | *Figures 1, 2, 4E* |
| pPW3405 | cytosolic GFP | pEGFP | KanR | expression of GFP | *Figures 1C-H* |
| pPW3406 | rhodopsin (RHO) -GFP | pRK | AmpR | expression of RHO-GFP | *Figure 1—figure supplements 4* and *5, Figure 2—figure supplement 1* |

*Continued on next page*

*Continued*

| Lab archive | Plasmid description | Vector | Resistance | Purpose | Construct used in figure(s): |
|---|---|---|---|---|---|
| pPW3407 | proinsulin (INS) -GFP | pRK | AmpR | expression of INS-GFP | *Figure 1—figure supplements 4 and 5* |
| pPW3426 | MPZ-ecto peptide-eGFP | pEGFP | KanR | expression of MPZ-ecto peptide (FTWRYQPEGGRDAISIFHYA) | *Figure 4* |
| pPW3427 | MPZ-ecto peptide$^{Tyr \to Glu}$-eGFP | pEGFP | KanR | expression of MPZ-ecto$^{Tyr->Glu}$ peptide (FTWREQPEGGRDAISIFHEA) | *Figure 4* |
| pPW3428 | MPZ-ecto peptide$^{Arom \to Glu}$-eGFP | pEGFP | KanR | expression of MPZ-ecto$^{Arom->Glu}$ peptide (ETEREQPEGGRDAISIEHEA) | *Figure 4* |
| pPW3429 | DR5 long FLAG-His6x, WT with silent mutation | pRK | AmpR | expression of DR5 long isoform-FLAG, harbors silent nt mutations within signal sequence to evade siRNA (AAGACCCTTGTGCTCGTTGTC → AAaACaCTTGTGCTCGTTGTC) | *Figure 4—figure supplement 1* |
| pPW3430 | CHOP (canonical isoform) | pRK | AmpR | expression of CHOP in absence of ER stress | *Figure 1—figure supplement 2* |

## Transfections with siRNA

For experiments shown in *Figure 1* and *Figure 2—figure supplement 1A*, the siRNA oligonucleotides against DR5 and a non-targeting control were purchased from Dharmacon (ON-TARGETplus Human TNFRSF10B 8795 siRNA # L-004448-00-0005 and ON-TARGETplus Non-targeting siRNA #2 # D-001810-02-05). The siRNA transfection was performed as previously described in (2) using Lipofectamine RNAiMAX.

For the knockdown in *Figure 4—figure supplement 1B*, we synthesized custom siRNAs from Dharmacon siRNA (DR5siRNA-2: 5' AAG ACC CTT GTG CTC GTT GTC UU 3', Nt siRNA: 5' AAA CCU UGC CGA CGG UCU ACC UU 3'). The siRNA transfection was performed 24 hr previous to the DR5L-FLAG and MPZ ecto-peptide-GFP plasmid co-transfection for a knockdown of 48 hr total.

## RNA extraction and generation of cDNA

Cells were grown in 6-well plates and harvested with 0.5 ml of TRIzol reagent (Life Technologies) per manufacturer's protocol to extract RNA. For cDNA synthesis, 500 ng of total RNA was reverse transcribed with 2 µl of the SuperScript VILO master mix (Life Technologies # 11755050) in a total reaction volume of 10 µl following the manufacturer's protocol for reaction temperature and duration. The reverse transcription product was diluted to 200 µl with 10 mM Tris-HCl pH 8.2 and used at 1:100 dilution for subsequent RT-PCR reactions.

## Semi-quantitative RT-PCR for *Xbp1* mRNA splicing

2 µl of cDNA was added to 0.2 µM of forward and reverse primers (Hs_XBP1_Fwd: 5' -GGAGTTAA-GACAGCGCTTGG- 3'; Hs_XBP1_Rev: 5' -ACTGGGTCCAAGTTGTCCAG-3'), 0.2 mM of each dNTP, 0.5 units of Taq DNA polymerase (Thermo Scientific). The reaction was set at an annealing temperature of 60.5°C with an extension time of 30 s for 26 cycles. The products were then visualized on a 3% agarose gel (comprised of a 1:1 mixture of low-melting point agarose and standard agarose) stained by 1:10000 SybrSAFE.

## Quantitative PCR

PCR samples were prepared as described by manufacturer's protocols from iQ SYBR Green Supermix (BioRad #17088800).

For each experiment, qRT-PCR reactions were set up in triplicate and run using a CFX96 Real Time System (Bio-Rad). Quantitation cycles were determined with CFX Manager 3.0 software (Bio-Rad) and then normalized to GAPDH as an internal control.

## Primers used for quantitative RT-PCR

| Gene | 5'-forward primer-3' | 5'-reverse primer-3' |
|---|---|---|
| MPZ | GGCCATCGTGGTTTACAC | GATGCGCTCTTTGAAGGTC |
| GFP | CTGACCTACGGCGTGC | CCTTGAAGAAGATGGTGCG |
| BiP (GRP78) | GTTCGTGGCGCCTTGTGAC | CATCTTGCCAGCCAGTTGGG |
| DR5 (TNFRSF10B) | TTCTGCTTGCGCTGCACCAGG | GTGCGGCACTTCCGGCACAT |
| CHOP (DDIT3) | AGCCAAAATCAGAGCTGGAA | TGGATCAGTCTGGAAAAGCA |
| GADD34 | GAGGAGGCTGAAGACAGTGG | AATTGACTTCCCTGCCCTCT |
| GAPDH | AGCCACATCGCTCAGACAC | TGGAAGATGGTGATGGGATT |

## Protein analysis by western blot

Media and cells for each sample were collected by cell scraping into cold PBS followed by a subsequent wash with 1 ml of cold PBS. The cell pellet was then resuspended in cold lysis buffer (30 mM HEPES pH 7.2, 150 mM NaCl, 1% Triton X100, 1x Roche protease inhibitor) and lysed via needle shearing on ice. Samples were centrifuged at 1000xg for 5 min, and the supernatants were mixed with sample buffer to a final concentration of 1% SDS, 62.5 mM Tris-HCl pH 6.8, 10% glycerol, 0.1% bromophenol blue, 50 mM DTT. Samples were boiled at 95°C and loaded on SDS-PAGE gels (Gen-Script). Samples were subsequently transferred onto nitrocellulose membranes, blocked with Odyssey buffer (Licor) for 1 hr at RT, and probed with primary antibodies diluted 1:1000 (unless otherwise specified) in Licor Odyssey buffer supplemented with 0.1% Tween 20% at 4°C overnight.

## Antibodies used for western blots

| Primary antibody | Species | Catalog # |
|---|---|---|
| anti-DR5 | rabbit | Cell Signaling Technology 8074 |
| anti-FADD | mouse | BD Biosciences 610400 |
| anti-GFP | mouse | Roche 11814460001 |
| anti-PARP | rabbit | Cell Signaling Technology 9542 |
| anti-caspase 8 5F7 | mouse | MBL International M032-3 |
| anti-caspase 3 | rabbit | Cell Signaling Technology 9662 |
| anti-GAPDH | rabbit | Abcam 9485 |
| anti-IRE1 14C10 | rabbit | Cell Signaling Technology 3294 |
| anti-Sec31A | mouse | BD Biosciences 612350 |
| anti-Sec23A | rabbit | Invitrogen PA5-28984 |
| anti-RCAS1 D2B6N XP | rabbit | Cell Signaling Technology 12290 |
| anti-Giantin | rabbit | Abcam ab24586 |
| anti-ERGIC53 | rabbit | Sigma Aldrich E1031 |
| anti-Fc | mouse | One World Lab #603–510 |
| anti-FLAG M2 | mouse | Sigma F1804 |
| anti-His 6x | mouse | Abcam ab15149 |

Bound primary antibodies were probed by HRP-conjugated secondary antibodies (Amersham, Piscataway, NJ, 1:10000) using enhanced chemiluminescence (SuperSignal; Thermo Scientific, Waltham, MA) detected by standard film or through the ChemiDoc XRS+ Imaging System (Bio-Rad).

## Caspase activity assay

Cells harvested from a 6-well plate were resuspended in 100 µl of lysis buffer (30 mM HEPES pH 7.2, 150 mM NaCl, 1% Triton X-100) and lysed via needle shearing (25G, 10 passes) followed by a 30 min incubation on ice. Supernatant was collected after centrifuging at 1000xg for 5 min. For the caspase activity assay, 10 µl of supernatant diluted with lysis buffer (1:25, 1:50, 1:100 to stay within linear range of luminescence measurements) was incubated with 10 µl of the luminogenic caspase glo 8 substrate (Promega #PRG8200) in a 384-white walled plate (Corning 3574) for 45 min at RT before measuring luminescence on a Spectramax-M5 plate reader.

## Flow cytometry staining for apoptosis analysis

Cells seeded in a 6-well plate were transfected with the specified plasmid for 24 hr and harvested via trypsinization in complete media to allow a 30 min recovery at 37°C. The samples were then centrifuged at 500xg for 5 min and resuspended In 150 µl of a 1:20 dilution of Annexin V-AlexaFluor 647 conjugate (Thermo Scientific #A23204). Samples were transferred to a 96-well U-bottom plate and incubated in the dark for 15 min at RT. The distribution of apoptotic cells was determined by flow cytometry on a BD LSR II.

## Trypan blue staining and quantification for cell death analysis

Cells seeded in a 6-well plate were transfected with the specified plasmid for 24 hr and harvested via trypsinization in complete media and ultimately resuspended in 400 µl of media to obtain a concentration of $1.0-3.0 \times 10^6$ cells/ml. The cell suspension was then mixed 1:1 with 0.4% Trypan blue (Sigma-Aldrich) and incubated at 37°C for 30 min. The percentage of cells staining positive for was then quantified using the Countess II FL Automated Cell Counter (ThermoFisher, catalog # AMQAF1000) default brightfield settings with disposable slides.

## Size-exclusion chromatography for cell lysates

Size-exclusion fractionation of cell lysate was performed as previously described in *Lu et al. (2014)* with minor modifications.

In a 15 cm dish, 4 million HCT116 cells were seeded 18 hr prior to transfection (see Transient transfections for protein expression). 24 hr post-transfection, cells were harvested by collecting all media (to collect detached, dying cells) and scraping the dish in 3 ml of cold PBS. The cell suspension was centrifuged at 500xg to pellet the sample. Cells were washed with an additional 1 ml of cold PBS, pelleted, and flash frozen in liquid $N_2$ prior to lysis.

Cells were resuspended in 600 µl of lysis buffer (30 mM HEPES pH 7.2, 150 mM NaCl, 1% Triton X-100) with 1x protease inhibitor (Roche) and lysed through a 25G needle with 10 passes. Lysates were clarified by centrifugation at maximum speed for 15 min at 4°C, and the supernatant (400 µl) was loaded onto a SuperDex200 10/300 GL column equilibrated with lysis buffer at a flow rate of 0.35 ml/min. Fractions were collected in 0.5 ml aliquots.

For the caspase 8 activity assay, 10 µl of each fraction was incubated with 10 µl of the caspase glo 8 substrate (Promega #PRG8200) in a 384-well white walled plate (Corning 3574) for 45 min at RT before measuring luminescence on a Spectramax-M5 plate reader.

For Western blots, every three fractions were pooled for a total of 1.5 ml and subjected to TCA precipitation to concentrate the protein content. Samples were then analyzed by SDS-PAGE and immunoblotted for GFP and DR5.

## DR5 immunoprecipitation (IP)

IP for DR5 was performed as previously described in *Lu et al. (2014)*, using anti-DR5 mAb 5C7-conjugated agarose beads gifted by David Lawrence of the Ashkenazi lab at Genentech Inc.

## GFP immunoprecipitation (IP)

15 cm dishes were seeded with 4 million HCT116 cells and allowed to recover for 18 hr prior to transfection. Samples were then transfected for 24 hr with the GFP-tagged ER trafficked protein, cytosolic GFP, or empty vector. To harvest apoptotic and living cells for each sample, all media and washes were collected. Cells were scraped in 2 ml of cold PBS and combined with the media and

washes. Samples were centrifuged to pellet cells, washed with 1 ml cold PBS, and flash frozen in liquid N$_2$ prior to lysis.

The cell pellets were resuspended in 750 μl of 30 mM HEPES pH 7.2, 150 mM NaCl, 1% Triton X-100 with 1x Roche protease inhibitor cocktail (if used subsequently for caspase glo 8 assay) or with 5x Roche protease inhibitor cocktail (if used for immunoblotting). The cells were lysed by mechanical shearing through a 25G needle for 13–15 plunges followed by incubation on ice for 30 min. The lysate was centrifuged at 2000xg for 5 min to remove debris and then incubated with 30 μl of GFP-Trap magnetic agarose beads (Chromotek gtma-20) for 4 hr at 4°C. The beads were then washed with 750 μl of lysis buffer for 10 min at 4°C for three rounds.

For measuring caspase activity, a fifth of the beads was resuspended in 40 μl of 30 mM HEPES pH 7.2, 150 mM NaCl, 1% Triton X-100. 10 μl of the resuspended mixture was incubated with 10 ul of the caspase glo 8 luminogenic substrate (Promega #PRG8200) for 45 min at RT before measuring luminescence.

For Western blots, samples were eluted by adding 35 μl of non-reducing SDS gel loading buffer (50 mM Tris-HCl (pH 6.8), 2% SDS, 0.1% bromophenol blue, 10% glycerol) and incubated for 10 min at 70°C. The beads were then magnetically removed from the eluted sample before adding DTT to a final concentration of 25 mM. The entire sample was loaded onto the gel and immunoblotted for DR5, caspase 8, and GFP, in that order.

## Immunofluorescence of DR5 and organelle markers in HCT116

HCT116 were seeded at 25000 cells per well in an 8-well glass-bottom μSlide (Ibidi 80827) pre-coated with Collagen IV (Sigma-Aldrich C6745, 0.03 mg/ml in PBS incubated for 30 min at RT, and then rinsed off with PBS x 3) in growth media 18 hr prior to transfection (see above for protocol). 24 hr post-transfection, the cells were fixed using 4% paraformaldehyde (PFA, Electron Microscopy Sciences) or methanol, depending on the antibody combination used for immunostaining (see table below). Samples were protected from light after fixation to preserve GFP fluorescence.

### Antibodies used for immunofluorescence

| Primary antibody | Species | Catalog # | IF dilution | Fixation method |
|---|---|---|---|---|
| anti-DR5 3H3 | mouse | Genentech | 1:100 | 4% PFA/methanol |
| anti-cleaved caspase 8 (Asp391) (18C8) | rabbit | Cell Signaling Technology 9496 | 1:50 | 4% PFA |
| anti-RCAS1 D2B6N XP | rabbit | Cell Signaling Technology 12290 | 1:200 | methanol |
| anti-Giantin | rabbit | Abcam ab24586 | 1:1000 | methanol |
| anti-ERGIC53 | rabbit | Sigma Aldrich E1031 | 1:100 | methanol |

For fixation with 4% PFA, 32% PFA was added directly to the well in growth media for a 1:8 dilution and incubated for 10–15 min at RT. Cells were then permeabilized with PHEM buffer (60 mM PIPES, 25 mM HEPES, 10 mM EGTA, 2 mM MgCl2-hexahydrate, pH 6.9) containing 0.1% Triton-X100 for 10 min at RT for three washes.

For fixation with methanol, media was removed and cells were washed with cold PBS. Then methanol (pre-cooled at −20°C) was added to the well and incubated at −20°C for 3–4 min. The methanol was promptly aspirated and replaced with PHEM buffer (without Triton X-100), followed by two more 10 min washes with PHEM at RT.

To block against non-specific antibody interactions, cells were rinsed in PHEM (without 0.1% Triton X-100) and then incubated with PHEM containing 2% normal goat serum (Jackson Immunoresearch Laboratories, 005-000-121) for 1 hr at RT. Primary antibodies were incubated in blocking solution overnight at 4°C at the dilution noted in the table above. Cells were then washed with PHEM buffer for 10 min at RT x three and incubated with secondary antibodies, anti-mouse-AlexaFluor546 (Invitrogen A11030) and anti-rabbit-AlexaFluor633 (Invitrogen A21071), diluted 1:1000 in PHEM with 2% normal goat serum. (Secondary antibody solutions were centrifuged at 15000xg for 20 mins at 4°C to remove aggregates prior to incubating with cell samples.) After a 1 hr incubation at RT, cells were washed with PHEM buffer once, then incubated with PHEM buffer containing

nuclear stain (DAPI, Molecular Probes, Eugene, OR, D-1306, 5 µg/mL), and finally rinsed with PHEM buffer twice before imaging.

Samples were imaged on a spinning disk confocal with Yokogawa CSUX A1 scan head, Andor iXon EMCCD camera, and 40x Plan Apo air Objective NA 0.95 or 100x ApoTIRF objective NA 1.49 (Nikon).

## Quantification of Pearson's correlation for co-localization analyses

Images were analyzed using CellProfiler (*Carpenter et al., 2006*). From the 405 nm channel, cell nuclei were identified as primary objects using maximum correlation thresholding (MCT) and shape to distinguish clumped objects. Secondary objects were outlined using propagation from the nuclei (DAPI) after applying MCT from the 488 nm, 561 nm, and 633 nm channels. Tertiary objects for each cell was identified as the mask of each nucleus subtracted from the mask of its corresponding secondary object in each channel, yielding masks for intracellular GFP-tagged protein from the 488 nm channel, for intracellular DR5 from the 561 nm channel, and for the organelle marker in the 633 nm channel. For overlap between DR5 and MPZ/RHO-GFP, Pearson's correlation coefficients were calculated within the MPZ/RHO mask for each cell to filter out untransfected cells. For overlap between DR5/MPZ and organelle markers, Pearson's correlation coefficients were calculated within tertiary objects for the organelle marker in GFP+ cells. Statistical analyses for all data sets were performed using GraphPad Prism 6.0.

## Subcellular fractionation

This subcellular fractionation protocol was adapted from *Xu et al. (2015)*.

Cells transfected with MPZ-GFP were harvested from a 15 cm dish as described in the Size Exclusion Chromatography section above. The cell pellet was resuspended in 400 µl of homogenization buffer (10 mM triethanolamine-acetic acid, pH 7.4, 0.25 M sucrose, 1 mM sodium EDTA, protease inhibitor cocktail (Roche)) and lysed via mechanical shearing through a 25-gauge needle on a 1 ml syringe with 13 passes. (Note on protease inhibitor: 4x protease inhibitor was used to lyse samples used for Western blotting to minimize degradation of proteins, while 1x protease inhibitor was used in sample lysis for caspase glo 8 assay so that excess inhibitor would not interfere with cleavage of luminescent substrate.) Homogenized sample was then centrifuged at 2000xg for 15 min at 4˚C to remove unlysed cells and the nuclear fraction. 250 µl of the supernatant was loaded via capillary action onto a 9–25% iodixanol gradient in a 13 × 51 mm polyclear centrifuge tube (Seton Scientific # 7022–29426) prepared using the BioComp Gradient Master (Model 107). The 9% and 25% layers of the gradient were made by diluting 60% iodixanol (OptiPrep Density Medium,Sigma Aldrich # D1556) into cell suspension medium (0.85% (w/v) NaCl, 10 mM Tricine-NaOH, pH 7.4). The loaded gradients were then centrifuged in SW55Ti rotor at 44800 rpm for 2 hr at 4˚C. After deceleration without braking, 300 ul fractions were collected from the top of the tube.

Fractions from samples lysed with 4x protease inhibitor were subjected to TCA precipitation to concentrate protein content for immunoblotting. Fractions from samples lysed with 1x protease inhibitor were used for the caspase glo 8 assay (10 µl of each fraction + 10 µl of the caspase glo 8 reagent (Promega #PRG8200) incubated for 45 min at RT prior to measuring luminescence).

## Peptide array binding

Peptide arrays were purchased from the MIT Biopolymers Laboratory as described previously in *Karagöz et al. (2017)* and *Gardner and Walter (2011)*. The sequences of each protein were tiled from N- to C-terminus in 18-amino-acid-long peptides shifting by three amino acids from the previous spot. The arrays were incubated in methanol for 10 min, and then washed in binding buffer (50 mM HEPES pH 7.2, 250 mM NaCl, 10% glycerol) for 10 min at room temperature x 3. 500 nM of Fc-tagged DR5 extracellular domain (ECD, a kind gift from Scot Marsters) was incubated with the array in binding buffer at room temperature for 1 hr. Then, the array was washed three times for 10 min with binding buffer to remove unbound protein. The bound DR5 ECD was then transferred onto a nitrocellulose membrane via a semi-dry transfer apparatus (Owl HEP-1) at 80 mA for 45 min at 4˚C. The membrane was then blocked with 1xPBST with 5% milk for 1 hr at room temperature and probed with 1:1000 anti-Fc (One World Lab #603–510) overnight at 4˚C followed by 1:10000 anti-mouse IgG-HRP (Promega #W4021) for 1 hr at RT. The membrane was imaged using chemi-luminescence with the Bio-Rad Universal Hood II Gel Doc below saturating pixel intensities. The signal of

each spot was quantified using Max Quant and normalized to the maximum intensity of all the spots on the same array.

## Peptides

Peptides were ordered from GenScript at >95% purity and stored with desiccant at −20°C. Peptides were dissolved as a highly concentrated stock solution in anhydrous DMSO and diluted 1:50 in aqueous buffer to measure the stock concentration using absorbance at 280 nm. All solution mixtures containing peptide and protein had a final concentration of 0.5% DMSO.

## Amino acid sequences of MPZ-derived peptides

| Peptide | Peptide sequence |
|---------|------------------|
| MPZ-ecto<sup>VD</sup> | VSDDIS<u>FTWRYQPEGGRD</u> |
| MPZ-ecto | <u>FTWRYQPEGGRD</u>AISIFHYA |
| MPZ-ecto<sup>Tyr→Glu</sup> | FTWREQPEGGRDAISIFHEA |

## Peptide pulldown with Fc-tagged ECD proteins

Recombinant Fc-tagged DR5 and TNFR1 ECD proteins were generated and purified at Genentech Inc by S. Marsters.

Fc-tagged proteins were incubated with Dynabeads Protein G (20 µg per 12.5 µl of beads for each sample) in 1xPBS for 1.5 hr at RT. Beads were then washed with 1xPBS via magnetic pulldown, followed by two 10 min washes with 20 mM HEPES pH 7.2, 100 mM KOAc, 0.2% Tween-20 at RT to remove unbound Fc-tagged protein. Protein bound-beads were incubated with 50 µl of 100 µM peptide in 20 mM HEPES pH 7.2, 100 mM KOAc, 0.2% Tween-20 for 1 hr at RT. Beads were then washed with 50 µl of buffer three times, and samples were eluted with 25 µl of non-reducing sample buffer (62.5 mM Tris-HCl pH 6.8, 2.5% SDS, 10% glycerol) by incubating at 65°C for 15 min to elute the complex.

## Purification of recombinant of DR5 and TNFR1 ECD

Human DR5 ECD (long isoform, residues 72–213: KRSSPSEGLCPPGHHISEDGRDCISCKYGQDYSTH WNDLLFCLRCTRCDSGEVELSPCTTTRNTVCQCEEGTFREEDSPEMCRKCRTGCPRGMVKVGDC TPWSDIECVHKESGTKHSGEVPAVEETVTSSPGTPASPCSLSG) and TNFR1 ECD (residues 22–211: I YPSGVIGLVPHLGDREKRDSVCPQGKYIHPQNNSICCTKCHKGTYLYNDCPGPGQDTDCRECESGSFTA SENHLRHCLSCSKCRKEMGQVEISSCTVDRDTVCGCRKNQYRHYWSENLFQCFNCSLCLNGTVHLSC QEKQNTVCTCHAGFFLRENECVSCSNCKKSLECTKLCLPQIENVKGTEDSGTT) were cloned into a pFastBac HTB vector containing a Gp67 (glycoprotein) signal peptide and 6xHis tag to force secretion of the expressed protein. The pFastBac HTB constructs were recombined into bacmid DNA using the Bac-to-Bac baculovirus expression system according to manufacturer's protocols (Life Technologies).

| Lab archive | Plasmid description | Vector | Resistance | Construct used in figure(s): |
|-------------|-------------------|--------|-----------|------------------------------|
| pPW3408 | Gp64-His6x-DR5 long ECD-FLAG | pFastBacHT | AmpR | 3D-3F |
| pPW3410 | Gp64-His6x-DR5 long ECD | pFastBacHT | AmpR | 3C |
| pPW3411 | Gp64-His6x-TNFR1 ECD | pFastBacHT | AmpR | 3C |

Table S6: Constructs used to generate bacmid for recombinant protein purification.

SF21 were grown in SF-900 II media supplemented with 10% FBS at 28°C in disposable Erlenmeyer flasks rotating at 150 rpm and transfected using Cellfectin II (Thermo Fisher Scientific) according to manufacturer's protocols. The baculovirus was amplified two more times at a low M.O.I. prior to infection of SF21 with a 1:50 dilution of the virus for protein expression.

After 72–96 hr of infection, the SF21 suspension was centrifuged at 2000xg for 15 min (two rounds) to collect the media containing the secreted ECD protein. The media was then further

clarified through a 0.2 um filter before loading onto a HisTrapFF column equilibrated with 25 mM imidazole pH 7.4, 150 mM NaCl (Buffer A) at a flow rate of 3.0 ml/min. The column was washed with 20 CV of Buffer A at a flow rate of 4.0 ml/min. To elute, the concentration of imidazole was increased through a linear gradient from 0–100% of Buffer B (500 mM imidazole pH 7.4, 150 mM NaCl) in 7 CV. Fractions containing the His-tagged ECD were then concentrated and further purified on a SuperDex200 10/300 GL column (GE Healthcare) equilibrated with 30 mM HEPES pH 7.2, 150 mM NaCl. For long-term storage, the protein was diluted into 30 mM HEPES pH 7.2, 150 mM NaCl, 10% glycerol and flash frozen in liquid $N_2$ before storing at −80℃.

### Labeling of DR5 and TNFR1 ECD

Recombinant DR5 and TNFR1 ECD were labeled with AlexaFluor647 NHS Ester (Succinimidyl Ester) (Life Technologies # A37573) in a 3:1 dye:protein molar ratio in 30 mM HEPES pH 7.2, 150 mM NaCl, 1% DMF overnight at 4℃ protected from light. The labeled proteins were re-loaded onto a SuperDex200 10/300 GL column to remove the excess dye, yielding labeling efficiencies of 59% and 49% for DR5 ECD and TNFR1 ECD, respectively. Labeled proteins were diluted into buffer with 10% glycerol and flash frozen in liquid $N_2$ for long-term storage at −80℃.

### Fluorescence quenching assay

To set up reactions, the unlabeled peptide titration (10 μl each) was made from a two-fold dilution series of the highest peptide concentration assayed. To each peptide sample, 10 μl of Alexa-Fluor647-labeled ECD protein was added to give a final concentration of 200 nM ECD protein in 20 mM HEPES pH 7.2, 100 mM KOAc, 0.1% Tween-20. Samples were incubated for 30 min at RT protected from light before measuring fluorescence.

Samples were then loaded by capillary action onto premium capillaries (Nanotemper Technologies, cat. #MO-K025). Fluorescence was measured on a Monolith NT.115 Instrument (NanoTemper Technologies, Germany) using the capillary scan function at 25℃.

The half-maximum binding constant ($K_{1/2}$) and Hill coefficient were determined by fitting the data points on Prism 6.0 to the model equation: $Y = B_{max}*X^h/(K_{1/2}^h + X^h)$, where Y is the % quenching, X is the concentration of peptide, $B_{max}$ is the maximum % quenching, and h is the Hill slope.

### Crosslinking for DR5 ECD and peptide

To set up the peptide titration series, two-fold dilutions were made from the highest peptide concentration used into 20 mM HEPES pH 7.2, 150 mM NaCl. An equal volume of C-terminal FLAG-tagged DR5 ECD was added to each peptide sample or buffer alone for a final concentration of 10 μM DR5 ECD. Reactions were equilibrated at RT for 30 min prior to the addition of 100 μM BS3 for 20 min at RT. Excess crosslinker was quenched by adding Tris-HCl pH 7.4 to a final concentration of 100 mM and incubated for 15 min at RT before analysis by SDS-PAGE. The resulting gel was then transferred onto a nitrocellulose membrane (120 V, 2 hr for high MW species) and blotted with anti-FLAG to visualize discrete crosslinked species.

### Size exclusion of DR5 ECD and peptide complex

The specified concentration of protein and peptide were incubated overnight at 4℃. The following day, 200 ul of each sample was loaded onto a SuperDex200 10/300 GL column equilibrated with 30 mM HEPES pH 7.2, 150 mM NaCl at a flow rate of 0.5 ml/min. Fractions were collected in 1 ml aliquots.

For subsequent analysis of each fraction by SDS-PAGE, 1 ml fractions were concentrated to 50 μl using a 3 kDa MW cutoff and loaded onto a gel. The fluorescein-labeled peptide was visualized using the fluorescence detection mode on a Typhoon 9400 Variable Mode Imager (GE Healthcare). Protein was visualized by staining with Coomassie.

## Acknowledgements

We thank D Lawrence at Genentech for gifting anti-DR5 5C7-conjugated agarose beads and helpful discussions. We are grateful to CM Gallagher for her detailed immunofluorescence protocols, to GE Karagöz for her insight on peptide binding assays, to NW Frankel for his help in flow cytometry, to K

Crotty for her quantitative PCR protocol, and to S Mukherjee for her thoughtful feedback. M Lam was funded by a National Science Foundation Graduate Research fellowship. S Marsters is Principal Staff Researcher and A Ashkenazi is Senior Staff Scientist of Genentech, Inc P Walter is an investigator of the Howard Hughes Medical Institute.

## Additional information

### Competing interests
Scot A Marsters, Avi Ashkenazi: Affiliated with Genentech Inc. The author has no other competing interests to declare. The other authors declare that no competing interests exist.

### Funding

| Funder | Grant reference number | Author |
| --- | --- | --- |
| National Science Foundation | Graduate Research Fellowship | Mable Lam |
| Howard Hughes Medical Institute | | Peter Walter |

The funders had no role in study design, data collection and interpretation, or the decision to submit the work for publication.

### Author contributions
Mable Lam, Conceptualization, Resources, Data curation, Formal analysis, Validation, Methodology; Scot A Marsters, Resources, Methodology; Avi Ashkenazi, Conceptualization, Resources, Supervision, Project administration; Peter Walter, Conceptualization, Resources, Supervision, Funding acquisition, Project administration

### Author ORCIDs

Mable Lam https://orcid.org/0000-0001-7016-2257
Avi Ashkenazi https://orcid.org/0000-0002-6890-4589
Peter Walter https://orcid.org/0000-0002-6849-708X

### Decision letter and Author response
Decision letter https://doi.org/10.7554/eLife.52291.sa1
Author response https://doi.org/10.7554/eLife.52291.sa2

## Additional files

### Supplementary files
• Transparent reporting form

### Data availability
All data have been reported in the manuscript and supporting files. Source data files have been provided in all figures.

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
