## [Decision Letter]

**Acceptance summary:**

The work of this group demonstrates a direct mechanism whereby misfolded or accumulated ER client proteins can elicit apoptosis through direct engagement of DR5. The work suggests a provocative model to account for how UPR signaling is necessary but not sufficient to elicit cell death, and how cells might tune their survival decisions to the strength and persistence of the stressor. It will be interesting in future work to see how DR5 distinguishes substrates, and to identify what other stress signaling pathways impinge on this pathway to regulate its output. I expect that this work will be of great interest to those interested in stress responses and cell fate decisions.

**Decision letter after peer review:**

Thank you for submitting your article "Misfolded proteins bind and activate death receptor 5 to induce apoptosis during unresolved endoplasmic reticulum stress" for consideration by *eLife*. Your article has been reviewed by three peer reviewers and the evaluation has been overseen by a Reviewing Editor and Vivek Malhotra as the Senior Editor. The following individual involved in review of your submission has agreed to reveal their identity: Cristina Muñoz-Pinedo (Reviewer #3).

The reviewers have discussed the reviews with one another and the Reviewing Editor has drafted this decision to help you prepare a revised submission.

Summary:

In this manuscript, the authors continue their studies on ER stress-induced cell death mediated by DR5. They demonstrate that overexpressed ER client proteins (primarily MPZ, but also rhodopsin) interact with DR5 and its partners (FADD, Casp8) in an ERGIC-like compartment. DR5 is capable of binding promiscuously and fairly weakly to peptides in MPZ and RHO, and expressing a wild-type MPZ peptide elicits death in a DR5-dependent manner while a mutant peptide that does not interact with DR5 does not (or at least not to as great an extent). Demonstration that the interaction can be broken such that approximately equal expression of the wild-type versus mutant peptide results in different outcomes is an important finding and helps raise confidence in the specificity of the interaction. The authors' model is appealing because it proposes that there is a mechanism by which the death pathway relies on some direct sensing of persistent misfolded proteins, rather than on the execution of a signaling cascade by UPR activation. There was agreement among the reviewers that the work is interesting and provocative. However, the reviewers also agreed that the paper requires a few straightforward experiments and some textual revisions to better support the conclusions and increase the rigor.

Essential revisions:

1) Ecto peptide studies. While the ecto peptide studies in Figure 4 add mechanistic strength to the paper, it is important to validate that the assumptions underlying the results are valid. Thus, (1) it is necessary to test whether the wild-type and mutant ecto peptides also accumulate in an ERGIC-like compartment with DR5. The data on wild-type full-length MPZ show that MPZ accumulation in this compartment is upstream of DR5 accumulation there. Therefore, if the mutant peptides do not accumulate in the ERGIC, it would suggest that their failure to elicit death is due to some other mechanism. Likewise if not even the wild-type peptide accumulates in the ERGIC, that would suggest an indirect DR5-dependent death mechanism. And (2) it is necessary to confirm that the cell death is in fact apoptotic (ideally by AnnexinV/PI and by showing that caspases are necessary, or both).

2) Increased Rigor. There were instances in which the reviewers were not fully confident in the robustness of the quantitative data in the manuscript. This hesitancy arose from instances of quantitative data where statistical comparisons were not made, and questions as to whether replicates were technical rather than biological in nature. To remedy this, all quantitative data should include appropriate statistical analysis including corrections for multiple comparison where appropriate and specification of the biological "n" number from which the data are derived (obviously if there are any data in which technical replicates rather than biological were used, biological replicates would need to be performed). Co-IP efficiencies should also be quantified for Figures 1 and 4. Particularly for Figure 1, the inputs in Figure 1—figure supplement 3B show equal DR5 in the three samples, but this is inconsistent with other experiments in the paper showing increased DR5 expression when MPZ is overexpressed. Thus, it is difficult to interpret increased pulldown of DR5 et al. in Figure 1F in light of this discrepancy, and these data are key to the paper. More information is needed on the timing of the pulldown and why DR5 total expression is not higher, and there should be a quantification of the percent recovered from all conditions from biological replicates. Source data should also be provided for quantitative experiments.

3) Consistency of data. This concern can be addressed textually. Not every assay is carried through every experiment in the paper (for instance, MPZ, FADD, caspase 8, and DR5 are not always IP'd together), and there are some results that seem on their face inconsistent with the model. For example, insulin elicits death in a DR5-dependent manner to as great or greater a degree as MPZ and RHO but does not appear to interact with DR5 or caspase 8 to a great extent (Figure 1—figure supplement 4D, E; Figure 1—figure supplement 5A, B). This alone suggests there are likely to be other factors and pathways involved. The system is likely to be complicated with many moving parts, and so the discrepancies here-and-there do not necessarily nullify the model, but these discrepancies should be explicitly acknowledged rather than minimized.

---

## [Author Response]

Essential revisions:1) Ecto peptide studies. While the ecto peptide studies in Figure 4 add mechanistic strength to the paper, it is important to validate that the assumptions underlying the results are valid. Thus, (1) it is necessary to test whether the wild-type and mutant ecto peptides also accumulate in an ERGIC-like compartment with DR5. The data on wild-type full-length MPZ show that MPZ accumulation in this compartment is upstream of DR5 accumulation there. Therefore, if the mutant peptides do not accumulate in the ERGIC, it would suggest that their failure to elicit death is due to some other mechanism. Likewise if not even the wild-type peptide accumulates in the ERGIC, that would suggest an indirect DR5-dependent death mechanism.

To determine if the peptides accumulate in the ERGIC, we have imaged the localization of GFP-tagged MPZ-ecto peptide and its Glu-containing mutants. All three plasma membrane-targeted constructs showed intracellular puncta that partially overlapped with ERGIC and DR5 signal (Figure 4F, Figure 4—figure supplement 1C-D). Quantification of the mean Pearson’s correlation with ERGIC or with DR5 for each MPZ-ecto peptide showed no significant difference.

Additionally, DR5 exhibits a positive correlation with ERGIC for all three conditions, despite the difference in co-IP enrichment and measured cell death. This indicates that retention of DR5 at the ERGIC occurs independently of its interaction with these trafficking substrates and may instead occur due to a general backlog of trafficking through the secretory pathway during ER stress.

And (2) it is necessary to confirm that the cell death is in fact apoptotic (ideally by AnnexinV/PI and by showing that caspases are necessary, or both)

To confirm that cell death occurs through apoptosis, we have now measured Annexin V and Sytox Blue staining for cells transfected with the MPZ-ecto peptides. Figure 4—figure supplement 3B-G shows the distribution of early (Annexin V+, SytoxBlue-) versus late (Annexin V+, Sytox Blue+) apoptotic cells for one biological replicate, and Figure 4H summarizes the percentage of cells positive for Annexin V averaged over three biological replicates. As expected, the MPZ-ecto peptide exhibits a substantially higher population of Annexin V+ cells. Furthermore, the addition of the pan-caspase inhibitor z-VAD reduced Annexin V staining (Figure 4H, Figure 4—figure supplement 3D), demonstrating the requirement of caspase activity.

2) Increased Rigor. There were instances in which the reviewers were not fully confident in the robustness of the quantitative data in the manuscript. This hesitancy arose from instances of quantitative data where statistical comparisons were not made, and questions as to whether replicates were technical rather than biological in nature. To remedy this, all quantitative data should include appropriate statistical analysis including corrections for multiple comparison where appropriate and specification of the biological "n" number from which the data are derived (obviously if there are any data in which technical replicates rather than biological were used, biological replicates would need to be performed).

We have now clarified biological versus technical replicates in the figure legends. For key experiments, we have stated exact p values and statistical analysis methods in the figure legends. Additional biological replicates confirmed the original conclusions of the results.

Specifically, we incorporated an additional biological replicate to Figure 1G (previously n = 2 biological replicates, with each containing three technical replicates, but now n = 3 biological replicates).

Figure 4C shows the average caspase activity of n = 3 biological replicates. We had originally shown the caspase activity (with technical replicates) for the full titration of ecto peptide expression shown in Figure 4B, but we decided to repeat this assay for the normalized expression levels of the peptides. The original technical replicates of the full titration have been provided in Figure 4—source data 1.

Figure 4E now includes two additional biological replicates of qPCR for MPZ-ecto peptide and its mutant variants, and statistical significance was determined using multiple t-tests with correction for multiple comparisons using the Holm-Sidak method.

We did not perform additional biological replicates for data that had been repeated elsewhere in the manuscript. Specifically, Figure 1—figure supplement 1E shows a 2-fold change of caspase activity after 24 hours of MPZ-GFP transfection, which agrees with Figure 1—figure supplement 3C showing an average 2-fold increase from three biological replicates. In Figure 4B, the quantification of the percent PARP cleaved from titrating ecto peptide expression also agrees with the quantification shown in Figure 4G for the selected expression levels of ecto peptide and its mutants.

We also did not repeat experiments for data that do not make new arguments. In particular, Figure 1—figure supplement 1A agrees with the well-established notion that increased expression of ER-trafficked proteins increases the transcription of UPR targets.

Co-IP efficiencies should also be quantified for Figures 1 and 4. Particularly for Figure 1, the inputs in Figure 1—figure supplement 3B show equal DR5 in the three samples, but this is inconsistent with other experiments in the paper showing increased DR5 expression when MPZ is overexpressed. Thus, it is difficult to interpret increased pulldown of DR5 et al. in Figure 1F in light of this discrepancy, and these data are key to the paper. More information is needed on the timing of the pulldown and why DR5 total expression is not higher […]

All pulldowns were performed with cells harvested 24 hours post-transfection when increased levels of DR5 were observed. We think that the saturation of the DR5 signal in the blot of Figure 1—figure supplement 3B may be obscuring the relative levels. To address this discrepancy, we repeated the IPs for MPZ-GFP to normalize the DR5 input levels to GAPDH (shown in Author response image 1). When normalized to GAPDH, the DR5 levels of MPZ-transfected cells were 2-fold higher. The GAPDH levels of the MPZ inputs were consistently lower than those of the GFP inputs because we obtained a lower yield of MPZ-transfected cells due to cell death.

**Author response image 1. respfig1:** Biological Replicates for MPZ-GFP Pulldown of DR5. (Left) Additional replicates of the MPZ-GFP IP from Figure 1G with signal for GFP and MPZ shown in green, GAPDH shown in red, and DR5 shown in grey; (Right) Quantification of DR5 signal in IP lanes normalized to GAPDH (n = 2 biological replicates, error bars are SD).

Nevertheless, even with less total DR5 in the input of MPZ samples, the MPZ IPs still showed higher amounts of DR5 than the GFP IPs.

[…] and there should be a quantification of the percent recovered from all conditions from biological replicates. Source data should also be provided for quantitative experiments.

We have now quantified the percent of DR5 recovered in pulldowns of the ER-trafficked candidates (Figure 1—figure supplement 5C). From three biological replicates, we estimate that MPZ-GFP retains 10% of DR5 from the input after three 10-min washes, while RHO-GFP retains 5%. The source data for this quantification is included in Figure 1—source data 9. This yield of DR5 is consistent with the weak affinity measured between DR5 and the MPZ-ecto peptide in solution. Since DR5 licenses caspase 8 to propagate downstream caspase signaling, only a fraction of DR5 would need to be activated to achieve apoptosis.

Likewise for Figure 4G, we estimate that 7-8% of total DR5 remains bound to the MPZ-ecto peptide from two biological replicates of IPs (Figure 4—figure supplement 3A). Both trials consistently show that DR5 is 3-4 fold more enriched on MPZ-ecto peptide than on the mutant peptides (See Figure 4—source data 4).

3) Consistency of data. This concern can be addressed textually. Not every assay is carried through every experiment in the paper (for instance, MPZ, FADD, caspase 8, and DR5 are not always IP'd together) […]

For the IPs of the MPZ-ecto peptide variants in Figure 4G, we did not show the immunoblot of caspase 8 because the abundance of the GFP-tagged peptides, which migrate at 40 kDa where the cleaved p43 subunit of caspase 8 would appear, interfered with the specificity of the caspase 8 antibody.

[…] and there are some results that seem on their face inconsistent with the model. For example, insulin elicits death in a DR5-dependent manner to as great or greater a degree as MPZ and RHO but does not appear to interact with DR5 or caspase 8 to a great extent (Figure 1—figure supplement 4D, E; Figure 1—figure supplement 5A, B). This alone suggests there are likely to be other factors and pathways involved. The system is likely to be complicated with many moving parts, and so the discrepancies here-and-there do not necessarily nullify the model, but these discrepancies should be explicitly acknowledged rather than minimized.

We have addressed the differences in DR5 engagement between MPZ, RHO, and INS in our Discussion section by pointing out the likelihood that overexpression of a singular protein (i.e. proinsulin) compromises the folding of other ER-folding proteins at endogenous levels, meaning that ER stress may cause a general misfolding of proteins that act as additional, and perhaps higher affinity, ligands for DR5.